# Functional Food Nutrients, Redox Resilience Signaling and Neurosteroids for Brain Health

**DOI:** 10.3390/ijms252212155

**Published:** 2024-11-12

**Authors:** Maria Scuto, Miroslava Majzúnová, Gessica Torcitto, Silvia Antonuzzo, Francesco Rampulla, Eleonora Di Fatta, Angela Trovato Salinaro

**Affiliations:** 1Department of Biomedical and Biotechnological Sciences, University of Catania, 95123 Catania, Italy; uni238065@studium.unict.it (G.T.); s.antonuzzo@gmail.com (S.A.); francescorampulla1985@virgilio.it (F.R.); 2Department of Animal Physiology and Ethology, Faculty of Natural Sciences, Comenius University, Ilkovicova 6, 84215 Bratislava, Slovakia; miroslava.majzunova@uniba.sk; 3Institute of Normal and Pathological Physiology, Centre of Experimental Medicine, Slovak Academy of Sciences, Sienkiewiczova 1, 81371 Bratislava, Slovakia; 4Oasi Maria S.S Research Institute—IRCCS, 94018 Troina, Italy; edifatta@oasi.en.it

**Keywords:** neurosteroids, functional food nutrients, polyphenols, neurohormesis, oxidative stress, neuroinflammation, Nrf2 pathway, GABA signaling, central nervous system disorders, cerebral organoids

## Abstract

The interplay between functional food nutrients and neurosteroids has garnered significant attention for its potential to enhance stress resilience in health and/or disease. Several bioactive nutrients, including medicinal herbs, flavonoids, and bioavailable polyphenol-combined nanoparticles, as well as probiotics, vitamin D and omega-3 fatty acids, have been shown to improve blood–brain barrier (BBB) dysfunction, endogenous neurosteroid homeostasis and brain function. These nutrients can inhibit oxidative stress and neuroinflammation, which are linked to the pathogenesis of various neurological disorders. Interestingly, flavonoids exhibit dose-dependent effects, activating the nuclear factor erythroid 2–related factor 2 (Nrf2) pathway at the physiological/low dose (neurohormesis). This leads to the upregulation of antioxidant phase II genes and proteins such as heme oxygenase-1 (HO-1) and sirtuin-1 (Sirt1), which are activated by curcumin and resveratrol, respectively. These adaptive neuronal response mechanisms help protect against reactive oxygen species (ROS) and neurotoxicity. Impaired Nrf2 and neurosteroid hormone signaling in the brain can exacerbate selective vulnerability to neuroinflammatory conditions, contributing to the onset and progression of neurodegenerative and psychiatric disorders, including Alzheimer’s disease, anxiety and depression and other neurological disorders, due to the vulnerability of neurons to stress. This review focuses on functional food nutrients targeting Nrf2 antioxidant pathway and redox resilience genes to regulate the neurosteroid homeostasis and BBB damage associated with altered GABAergic neurotransmission. By exploring the underlying molecular mechanisms using innovative technologies, we aim to develop promising neuroprotective strategies and personalized nutritional and neuroregenerative therapies to prevent or attenuate oxidative stress and neuroinflammation, ultimately promoting brain health.

## 1. Introduction

The exploration of the molecular mechanisms underlying the interaction between neurosteroids and functional food nutrients in health and/or disease has gained considerable attention in the scientific community. The central nervous system (CNS) is strongly influenced by steroid hormones. Circulating steroid hormones, such as those released by the adrenal glands, placenta and gonads, are lipophilic and readily cross the BBB to reach the brain, where they are known as neuroactive steroids [1]. However, neurosteroids can also be synthesized de novo by the nervous system from cholesterol and display beneficial neuroprotective properties [2]. Neurosteroid hormones play pivotal roles in cellular resilience response, immune response, neuroinflammation, synaptic adaptations, neurotransmitter regulation, and genetic and epigenetic homeostasis [3]. In humans, neurosteroids, particularly dehydroepiandrosterone (DHEA), its sulfate ester (DHEAS), pregnenolone and allopregnanolone, can be found in higher concentrations in the brain than in the serum [4,5]. Notably, abnormalities in the synthesis and function of neurosteroids have been closely linked to the onset and progression of nervous system disorders such as Alzheimer’s disease (AD), Parkinson’s disease (PD), schizophrenia (SZ), epilepsy, autism spectrum disorders (ASD), depression and brain tumors [6,7,8]. Given their fundamental role in brain function and pathophysiology, neurosteroids, particularly in synergistic combination with nutritional strategies targeting Nrf2 and phase II antioxidant genes, could provide potential preventive and therapeutic approaches. By exerting a greater neuroprotective effect, these combined strategies can counteract oxidative damage and neuroinflammation, which in turn can lead to changes in excitatory and inhibitory neurotransmitters and the onset of a wide range of degenerative and neuropsychiatric diseases [9,10]. Oxidative stress is a biological redox mechanism implicated in the deregulation of neurosteroid hormone signaling, leading to various neuropathological conditions. It arises from an imbalance between reactive oxygen species (ROS) produced by mitochondria and the available antioxidant capacity. Excessive ROS can damage RNA, DNA, proteins and lipids, promoting cell apoptosis, senescence, BBB dysfunction, and telomere shortening, ultimately leading to nervous system disorders [11]. However, a physiological level of oxygen/nitrogen free radicals and non-radical reactive species (ROS/RNS) is termed oxidative eustress or “good stress.” This is characterized by low to mild levels of the pro-oxidants involved in the control of multiple biochemical processes, such as carboxylation, hydroxylation, peroxidation, and the modulation of signal transduction pathways like nuclear factor-κB (NF-κB), phosphoinositide-3-kinase and mitogen-activated protein kinase (MAPK) cascade. These processes contribute to brain health and neurosteroid homeostasis by regulating neurotransmitters like glutamate, gamma-aminobutyric acid (GABA), dopamine, serotonin, and N-methyl-D-aspartate (NMDA) receptors [11,12]. Conversely, excessive levels of ROS/RNS, generated from both endogenous (mitochondria, NADPH oxidases) and exogenous sources (radiation, certain drugs, foods, cigarette smoking and environmental pollutants), cause a harmful condition described as oxidative stress or “bad stress.” To adapt to continuous environmental stressors, cells, particularly neurons, have evolved a complex antioxidant redox system consisting of enzymatic antioxidant molecules, such as superoxide dismutase (SOD), catalase (CAT), glutathione peroxidase (GPX) and thioredoxin (Trx) to counteract excessive stress and inflammation in the brain [13,14,15,16,17]. From this new perspective, functional food nutrients, including phenols, flavonoids, carotenoids, terpenoids and vitamins as well as omega-3 and omega-6 polyunsaturated fatty acids (ω-3/ω-6 PUFAs), can activate low molecular antioxidants such as glutathione (GSH), hydrogen sulfide and carnosine and especially redox resilience genes and proteins such as heat shock protein 70 (Hsp70), γ-glutamyl cysteine synthase (γ-GCs) and sirtuins (Sirt1-6). The concerted action of these molecules helps balance oxidative stress and can prevent and treat aging and several chronic nervous system disorders [18,19,20]. Compelling evidence has revealed that GSH depletion in the left dorsolateral prefrontal cortex leads to oxidative damage, with higher levels of estradiol and total testosterone in females, which may be related to psychiatric and neurodegenerative disorders [21]. Neurosteroids exhibit neuroprotective and anti-neuroinflammatory effects primarily through their interaction with GABA_A_ receptors (GABARs) in various brain injury models [22,23,24,25]. In line with this, recent evidence has shown, in a rodent model of intracerebral hemorrhage, that 17β-estradiol provides neuroprotection by restoring BBB integrity and suppressing oxidative stress through the PI3K/Akt pathway [22]. In addition, recent evidence suggests that the neurosteroid 5-androstenediol inhibits oxidative injury by increasing SOD activity and decreasing malondialdehyde (MAD) levels. It also reduces inflammatory mediators such as toll-like receptor 4 (TLR4), NF-κB and high mobility group box 1 (HMGB1), as well as collagen deposition markers like transforming growth factor beta 1 (TGFβ1) and alpha smooth muscle-actin (α-SMA) levels [23]. Furthermore, a natural plant-derived neurosteroid called disogenin has been shown to exert neuroprotective effects in the hippocampus, by improving spatial learning and memory and protecting animals against the amyloid-β (1-42) peptides that disrupt cognitive functions, after 28 days of treatment in a dose-dependent manner [24]. Deregulation of physiological levels of neurosteroids or inhibition of their activity due to increased environmental stressors or pollutants poses a serious threat to brain health. Functional foods, which are natural or processed foods containing biologically active compounds, can offer potential benefits when consumed at appropriate dosages. These compounds can activate antioxidant pathways and have been clinically proven to prevent, manage, or treat a wide range of chronic disorders, including those affecting the nervous system [25]. Modulating neurosteroid levels through the consumption of functional food nutrients, including polyphenols, polyphenol-combined nanoparticles, vitamin D, probiotics and other natural neuroprotective molecules, such as ω3 and ω9 unsaturated fatty acids (α-linolenic acid (ALA), eicosapentaenoic acid (EPA), docosahexaenoic acid (DHA), and oleic acid (OA)), can activate multiple cellular and molecular pathways to enhance brain resilience and neuronal adaptive responses. These nutrients can help prevent or attenuate oxidative stress, BBB dysfunction, and neuroinflammation, thereby mitigating perturbations in neurosteroid signaling and the progression of CNS disorders [26,27,28]. Emerging evidence indicates that flavonoids, like genistein, daidzein, hesperetin, apigenin, naringenin and eriodictyol, can significantly decrease deoxycorticosterone and androstenedione levels by inhibiting 3β-hydroxysteroid dehydrogenase. Nevertheless, apigenin has been shown to be more potent than other polyphenols in raising the levels of pregnenolone and 17α-hydroxyprogesterone, thereby inhibiting cytochrome P450 (CYP) 17, CYP21, and 3β-hydroxysteroid dehydrogenase [28]. Additionally, apigenin and taxifolin have been found to inhibit neurosteroidogenic enzymes, like 5α-reductase 1 (SRD5A1), 3α-hydroxysteroid dehydrogenase (AKR1C9), and retinol dehydrogenase 2 (RoDH2), in cells and rats [28,29]. In neurons, stress-responsive sirtuins such as Sirt1 and Sirt3 act as cellular sensors that detect energy availability and modulate metabolic pathways. Intriguingly, neurons exposed to 10 μM triclosan in synergy with synthetic flavonoids exhibit increased expression of Sirt1 and Sirt3, which can inhibit the activation of estrogen receptors and regulate the androgen receptor by interacting with the aryl hydrocarbon receptor (AhR) [30]. This review discusses the neurohormetic effects of functional food nutrients alone and/or in synergy with neurosteroids, focusing on their ability to target the Nrf2 pathway, redox resilience genes, and GABAergic signaling to protect against neuroinflammation, oxidative stress, and BBB damage. Herein, we highlight the potential of neurosteroids, such as 17β-estradiol, DHEA, DHEAS, progesterone, and allopregnanolone, and their crosstalk with nutrients to explore novel anti-neuroinflammatory approaches and personalized nutritional and neuroregenerative therapies using innovative technologies in CNS disorders in order to promote brain health.

## 2. Neurohormesis and Food Nutrients for Brain Resilience

Neurohormesis is a neuroadaptive response to a moderate level of stress that enhances the nervous system’s ability to resist severe, potentially lethal stressors or those that could cause brain damage or disease. This adaptive phenomenon involves important mechanisms and pathways associated with brain resilience, including neurogenesis, synaptic plasticity, neurosteroid and neurotransmitter homeostasis, as well as tolerance to oxidative damage via activation of the Nrf2 pathway to prevent or slow the onset of neurodegenerative and neuropsychiatric diseases [31]. Brain resilience is conceptualized as a multifaceted neuronal process that enhances psychological adaptability to adversity, trauma, tragedy or significant sources of stress through adaptive mechanisms within the immunological system, brain, hypothalamo–pituitary–adrenal (HPA) axis and autonomic nervous system (ANS) axis and redox resilience pathways and genes. It represents the capacity of individuals to avoid conditions like depression and anxiety under stress, challenging the notion that extraordinary abilities or coping mechanisms are essential for stress adaptation. Therefore, identifying personalized biomarkers of brain resilience through a nutritional approach can help restore impaired Nrf2 and neurosteroid signaling, allowing us to characterize biologically vulnerable individuals into resilient individuals, such as centenarians, who serve as models of longevity and healthy aging [32]. In neurofunctional concert with peptides and neurotransmitters, the end products of the HPA axis, glucocorticoid and neurosteroid hormones, promote brain resilience and neurohormetic responses to environmental challenges [33,34]. Neurohormetic nutrients, including flavonoids, such as resveratrol, Hidrox^®^, sulforaphane, curcumin, 3-epigallocatechin gallate, but also vitamin D, probiotics and ω3 fatty acids, protect neurons against injury and disease in a dose-dependent manner. Notably, a sub-toxic dose of functional food nutrients such as flavonoids and their metabolites, particularly polyphenol nanoparticles, can induce neurohormesis, triggering adaptive neuronal stress responses driven by the modulation of phase II resilience genes and proteins, such as Hsp70, HO-1, GSH, histone deacetylases of the sirtuin family, lipoxin A4 (LXA4), neurotrophic factors such as brain-derived neurotrophic factor (BDNF) and transcription factors of the forkhead (FOXO) family. In addition, carotenoids (e.g., β-carotene and lycopene), ascorbic acid, tocopherols, ω3 fatty acids and probiotics can also activate endogenous molecules (e.g., bilirubin, hydrogen sulfide and carnosine), proteins (e.g., ferritin and transferrin) and antioxidant enzymes (e.g., SOD, GPX and catalase). These molecules effectively scavenge ROS/RNS, providing neuroprotection in a variety of cells, including neurons, thus preserving brain health in the face of a stressor or disease, as observed in recent preclinical and clinical studies (Figure 1) [35,36,37,38,39,40,41].

The neuroprotective actions of various functional food nutrients are explicated through reduced levels of oxidative damage and neuroinflammation, which protect cultured neurons against ROS-mediated cell apoptosis [42]. For example, rutin and quercetin might exert a neurohormetic response involving many signaling pathways and molecular networks associated with the modulation of proteasome function [43]. Importantly, a low dose of curcumin induces neurohormetic mechanisms by upregulating the resilience gene HO-1 to counteract oxidative and nitrosative stress during neurodegenerative disorders. On the other hand, the overactivation of HO-1 may have toxic effects on neurons [44]. Conversely, in cancer, high concentrations of curcumin promote pro-oxidant responses, leading to mitochondrial destabilization due to calcium release from the endoplasmic reticulum, autophagy, inhibition of invasion and metastasis, thus enhancing the efficacy of current chemotherapeutics [45]. Furthermore, trials have demonstrated the neurohormetic effects of a subtoxic dose (1 mg/kg) of 5,8-dihydroxy-1,4-naphthoquinone (naphthazarin) in PD models. The study revealed that, under neurodegenerative conditions, naphthazarin improved motor ability, prevented apoptosis of dopaminergic neurons and mitigated neuroinflammation. The neuroprotective effect of naphthazarin was mediated by inhibition of astroglial activation in response to 1-methyl-4-phenylpyridine (MPTP)-induced PD in a concentration-dependent manner [46]. In addition, an adequate uptake of ω-3/ω-6 PUFAs and astaxanthin provided neurological benefits in both animals and patients with neurodegenerative diseases [47]. Neurohormesis is a homeostatic approach of considerable importance and can be applied to the effects of neurosteroids to promote adaptation to neuronal stress across multiple biological endpoints. Importantly, neurosteroids follow a U-shaped neurohormetic dose–response curve, with low-dose effects stimulating neuronal adaptive responses and high-dose effects inhibiting them [48]. Indeed, it is well known that 17β-estradiol at higher doses causes downregulation of its receptors, which does not occur in the physiological concentration range [49]. Progesterone and its metabolites have been shown to induce both anxiogenic and anxiolytic effects depending on the endogenous dose. Paradoxically, an inverted U-shaped dose–response curve has been observed in animals treated with allopregnanolone and in postmenopausal women treated with progesterone. Specifically, low doses or concentrations increase anxiogenic effects, while high doses decrease these effects, displaying anxiolytic properties probably mediated via GABARs [50]. Overall, neurohormesis, supported by accurate physiological information from redox stress biomarkers and neurosteroid signaling may thus dictate novel neuropharmacological strategies to create specific natural formulations targeting the neuronal Nrf2 pathway and phase II resilience genes. These formulations could drive innovative personalized nutritional and neuroregenerative therapies for nervous system disorders. It is important to emphasize that dosage is a crucial factor in promoting neuroprotective or detrimental effects, especially of dietary nutrients and neurosteroids; therefore, dosage should be carefully evaluated.

## 3. Neurosteroidogenesis

Neurosteroidogenesis, the de novo synthesis of steroids by the nervous system, is a dynamic process modulated by a considerable number of endogenous factors. Fluctuations of neurosteroidogenesis occur under physiological conditions, such as brain development, ovarian cycle and pregnancy, as well as in neuropathological conditions like AD, PD, ASD, depression and brain tumors. Notoriously, the process begins in the mitochondria, where cholesterol is transported through the action of steroidogenic acute regulatory protein (StAR) and translocator protein 18 kDa (TSPO) [51]. Within the mitochondria, cholesterol is converted into pregnenolone by the mitochondrial side-chain cleavage enzyme (P450scc). The conversion of pregnenolone to progesterone leads to the synthesis of androgens and estrogens, for which several enzymes of the endoplasmic reticulum (ER) are required. For instance, the metabolism of progesterone to dihydroprogesterone (DHP) by 5α-reductase is found in oligodendrocytes and astrocytes [52], while 3α-hydroxysteroidoxireductase (3α-HSOR) is present in oligodendrocytes, astrocytes and neurons of the spinal cord [53] and metabolizes DHP into the potent GABAR agonist allopregnanolone [54]. Furthermore, aromatase is typically found in the ER of neurons but becomes highly expressed in astrocytes after brain injury or in AD pathogenesis [55]. In the following steps, pregnenolone is converted to progesterone or 17OH-pregnenolone. The latter is further converted to DHEA. Both pathways converge, as progesterone and DHEA are used for the synthesis of androstenedione and subsequently testosterone. Finally, testosterone is converted to 17β-estradiol by the aromatase enzyme [56]. Research shows that brain regions, particularly the cortex, hippocampus, hypothalamus and cerebellum, express high steroidogenic activity [57]. Overall, we postulate that modulation of neurosteroidogenesis through functional food nutrients restores altered endogenous neurosteroid levels and may represent a promising therapeutic approach by which to promote brain health in neurodegenerative and neuropsychiatric disorders.

## 4. The Interplay Between Functional Food Nutrients and Neurosteroids in Health and/or Disease

Functional food nutrients, including polyphenols i.e., flavonoids and non-flavonoids, such as phenolic acids, stilbenes, terpenes, phlorotannins and lignans, as well as vitamin D and ω3 fatty acids, possess preventive and pharmacological properties. These nutrients can modulate the deregulation of neurosteroid hormones by restoring their physiological levels in the brain [58,59].

### 4.1. Resveratrol

Resveratrol (RSV) is a polyphenolic phytoalexin found in numerous plants, including berries, grapes and nuts [60]. It exerts protective effects against metabolic, cardiovascular, neurodegenerative and psychiatric disorders [61]. Preclinical evidence from mice suggests that a low dose (4 mg) of RSV for 7 days reverses the cognitive decline in different types of memory (working, nonspatial, and locomotor functions) that is caused by lipopolysaccharide (LPS). Particularly, the authors showed that RSV significantly increased both estradiol and neprilysin levels to decrease Aβ deposition in the brain [62]. It is well known that RSV is an allosteric activator of the Sirt1 pathway in a dose-dependent manner. Consistent with steroid hormones, studies have reported that RSV modulates Sirt1 signaling via hormesis, indicating that it may evoke adaptive responses to stress. Notably, low doses of RSV (1 µM) enhance Sirt1 signaling and induce estrogen receptor-α (ERα) activation by mimicking estradiol, whereas maximal coactivation at 10–20 µM but a reduction at ≥50 µM has been also observed. Therefore, the intrinsic estrogenicity of RSV may underlie its proven ability to confer estradiol-mediated benefits and provide a mechanistic insight into how RSV delays aging and extends lifespan or protects against aging-related diseases by promoting healthy aging in vitro and in vivo [63]. Polyphenols present in grapes can influence the vital regulators of reproductive processes, particularly hypothalamic neurohormones (e.g., GnRH, oxytocin, LH and FSH), steroid hormones (e.g., progesterone, testosterone and estradiol) and prostaglandins [64]. Specifically, the flavanol myricetin found in red wine is able to block the release of insulin-like growth factor I (IGF-I)-induced progesterone by granulosa cells and to activate IGF-I-induced estradiol production [65]. Similarly, RSV in vitro enhances the secretion of prolactin and IGF-I binding protein 1 (IGFBP1), resulting in improved decidualization of human embryonic stem cells (ESCs) in a dose-dependent manner [66]. Concerning the hormetic effects on steroidogenesis, studies have shown that low doses (0.1 to 10 μg/mL) of grape seed extracts and proanthocyanidin B2 reduce ROS content in the cells, while increasing cyclin-dependent kinase inhibitors (CDKIs) p21 and p27 and decreasing cyclin D2. High doses (50–100 µM) have the opposite effect. Similarly, doses ranging from 0.1 to 100 μg/mL of grape seed extracts and proanthocyanidin B2 increased the release of estradiol and progesterone, along with higher levels of cholesterol carriers, cyclic adenosine monophosphate response element-binding protein (CREB), StAR and mitogen-activated protein kinases extracellular signal-regulated kinases 1/2 (MAPK/ERK1/2) phosphorylation in both primary human granulosa tumor cells and luteinized human granulosa cells [67]. Additionally, procyanidin B dimers in red wine and grape seeds can regulate aromatase, an inhibitor that suppresses in situ estrogen biosynthesis in breast cancer cells and xenograft mouse models [68]. Notably, RSV, at a dose of 10 μM, significantly inhibits the synthesis of the DHEA, androstenedione and 11-deoxicortisol activated by adrenocorticotropic hormone (ACTH) in fetal adrenocortical cells. These findings indicate that high doses of RSV in early pregnancy could have adverse toxic effects [69]. Moreover, RSV can competitively bind to the steroid site, directly inhibiting neurosteroidogenic enzymes, especially 5α-reductase 1 (SRD5A1), 3α-hydroxysteroid dehydrogenase (AKR1C9) and retinol dehydrogenase 2 (RDH2), thereby regulating local neurosteroid levels, including allopregnanolone and 5α-androstanediol in the rat brain [70]. Furthermore, a recent study has demonstrated that co-administration of L-carnosine (200 mg/kg b.w/day) and RSV (20 mg/kg b.w/day) significantly alleviates the toxic effects of alkylating drugs, such as busulfan, and promotes healthy effects for spermatogenesis by the recovery of both testis and sperm parameters in rats [71]. In addition, other studies have revealed that combined therapy with metformin (300 mg/kg/day) and RSV (20 mg/kg/day) improves hormone profile and ovarian follicular cell architecture by upregulating SIRT1 and AMPK in albino rats [72]. Importantly, similar to the drug sertraline (15 mg/kg), long-term administration of RSV (20 and 40 mg/kg) antagonized the decrease in progesterone and allopregnanolone levels in the prefrontal cortex and hippocampus, preventing or attenuating HPA dysfunction [73]. In humans, a high dose of RSV (1000 mg) for 4 months has been shown to significantly reduce serum levels of the androgen precursors androstenedione, DHEA and DHEAS in middle-aged men with metabolic syndrome (Table 1) [74].

### 4.2. Curcumin

Curcumin (CUR) is the active polyphenol derived from the turmeric plant, *Curcuma longa* [76]. Its neuroprotective effects, which involve penetrating the blood–brain barrier and diffusing into nervous tissue to improve neurosteroid deregulation, have been reported both in vitro and in vivo (Table 2) [77]. Recent preclinical evidence has demonstrated the neuromodulatory effects of CUR (100 mg/kg daily, orally) on depression-related neuroinflammation, comparing its effects to those of fluoxetine (FLX) and estradiol (E2) in ovariectomized rats. The results suggest that CUR increases cognitive performance and regulates dopamine and norepinephrine levels in different brain areas. Additionally, CUR negatively regulates monoamine oxidase B and positively regulates tyrosine hydroxylase in the limbic region. Furthermore, CUR significantly mitigates serum corticosterone hormone release, reduces the production of pro-inflammatory cytokines, and decreases malondialdehyde levels, leading to an increase in total antioxidant capacity in the limbic system [78]. Moreover, CUR (100 mg/kg, orally, once daily) attenuates boldenone-induced neurobehavioral disturbances, restores the oxidant/antioxidant balance, and represses the TLR4/MyD88/TRAF-6/NF-κB pathway and its pro-inflammatory signaling molecules such as TNF-α and IL-1β in rats [79]. A recent study has demonstrated that a synergistic treatment, combining glutamine, Curcuméga^®^ (a food supplement containing curcumin and polyunsaturated n-3 fatty acids) and Gabolysat^®^ (bioactive peptides from a fish hydrolysate) reduces colonic hyperpermeability and inflammatory markers (CXCL1, TNFα and IL1β) in murine models of irritable bowel syndrome (IBS) more effectively than each compound alone [80]. Additionally, a combined treatment with CUR (2.4 mg/kg) and lycopene (12.5 mg/kg) significantly attenuates prostate hyperplasia by blocking the upregulation of testosterone, 5α-dihydrotestosterone (DHT), 5α-reductase, estradiol (E2) and prostate-specific antigen (PSA) expression as well as the levels of inflammatory factors, including IL-1β, IL-6 and TNF-α, in rats [81]. Additionally, in ovarian polycystic syndrome (PCOS), CUR (200 mg/kg) exerts protective effects by modulating serum hormone levels (e.g., 17 β-estradiol, follicle stimulating hormone, luteinizing hormone, progesterone, and testosterone). Specifically, curcumin downregulates serum testosterone, insulin receptor substrate 1 (IRS1), phosphatidylinositol-3-kinase (PI3K) and protein kinase B (AKT) levels and upregulates glucose 4 (GLUT4) and phosphatase and tensin homolog (PTEN) expression in rats [82]. Furthermore, CUR (80 mg) reduces poor semen quality and restores testosterone hormone levels (3β-hydroxysteroid dehydrogenase and 17β-hydroxysteroid dehydrogenase) by repressing oxidative stress through a reduction in lipid peroxidation products and NF-κB expression and an increase in antioxidant enzymes including SOD, CAT and GPx levels in male rats [83]. Recent clinical trials have shown that patients with PCOS treated with CUR (500 mg three times daily) or placebo for 12 weeks experienced improvements in hyperandrogenemia and hyperglycemia with a reduction of DHEA and fasting plasma glucose (FPG) [84]. Finally, a recent pilot study reports that CUR (75 mg), in synergy with a 5α-reductase inhibitor, teopolioside (35 mg), improved symptoms associated with hyperandrogenism after 12 weeks [85].

### 4.3. Sulforaphane

Sulforaphane (SFN), an isothiocyanate derived from glucoraphanin and found in cruciferous vegetables like broccoli, possesses various health benefits, including anti-inflammatory, antioxidant and neuroprotective properties, with a hormetic dose–response profile (Table 3) [86]. Importantly, SFN exhibits preventive and therapeutic potential in regulating neurosteroid signaling both in vitro and in vivo [87,88]. Consistent with this, it has been reported that a low dose (10–50 nM) of 17β-estradiol enhances the ability of SFN to counteract oxidant stress by reducing intracellular ROS and 8-hydroxy-2′-deoxyguanosine (8-OHdG) levels and increasing the expression of phase II enzymes, such as GSH, HO-1, SOD, CAT, Trx and NQO1, after H_2_O_2_ exposure in cardiomyocytes [87]. Furthermore, low doses of SFN (10 mg/kg) significantly enhance 3α-hydroxysteroid dehydrogenases (3α-HSD) in mouse liver and murine hepatocytes, and reduces testosterone and DHT concentrations in plasma [88]. It is noteworthy that Sirt1, an enzyme that deacetylates transcription factors, has been shown to be associated with the cellular resilience response under oxidative stress. Based on this, a recent study by Chung et al. has demonstrated that cytoprotective Sirt1 and/or Nrf2 pathways are implicated in regulating the oxidant/antioxidant environment of Leydig cells, and thus in testosterone production. In particular, the authors report that these protective pathways are present in Leydig cells. Interestingly, low doses of 1 μM of honokiol in synergy with SFN increased Sirt1 and Nrf2 levels, resulting in the control of testosterone homeostasis despite the exposure of cells to oxidative agents [89]. More recently, a study revealed that low doses of 1 or 10 μM of SFN, when combined with vitamin D, exerted powerful anti-inflammatory activity in age-related macular degeneration. Specifically, the authors have demonstrated that this combined treatment lowered ROS production and the expression of pro-inflammatory mediators in vitro [90]. In addition, SFN and vitamin D, in synergy, have a potential application in prostate cancer therapy via the modulation of the JNK/MAPK signaling pathway [91]. Of note, SFN, progesterone and lipoic acid have recently been considered neuroprotective candidates in retinal degeneration, as their synergistic combination effectively decreases oxidative stress and increases the activation of antioxidant enzymes, including GSH, GPx and SOD2 [92].

### 4.4. Hidrox^®^

Hidrox^®^ (HD) is a freeze dried phenolic constituent in powder extract that derives from the aqueous fraction of olives that is obtained from defatted olive pulps during the processing of Olea europaea L. after olive oil extraction [93]. Compelling evidence has shown that a 12% HD extract contains several antioxidants with great therapeutic potential. In HD the most relevant polyphenol is represented by hydroxytyrosol (HT) (40–50%), followed by oleuropein (5–10%), which is approximately 20% oleuropein aglycone and gallic acid and 0.3% tyrosol [94]. Recent preclinical studies have shown that low doses (10 mg/kg) of HD exhibit antioxidant and anti-inflammatory effects by upregulating Nrf2 signaling and redox resilience genes such as HO-1 as well as by downregulating the neuroinflammation mediated by NF-κB signaling and proinflammatory cytokine cascade such as IL-1β, IL-6 and TNF-α, in order to prevent or slow the neurodegenerative process that is typical of AD and PD (Table 4) [95,96,97]. The activation of the Nrf2/HO-1 pathway restores GSH, SOD and catalase in both the bladder and spinal cord of rodents [97]. Interestingly, our recent studies in *Caenorhabditis elegans* have demonstrated that moderate/low concentrations of 250 mg of HD, when compared with synthetic HT, display greater neuroprotective action by increasing lifespan and stress resistance and by reducing neurotoxic aggregates of misfolded α-synuclein in dopaminergic neurons of transgenic PD models [98,99]. In the brain, HT is present as a product of dopamine metabolism. Importantly, 3,4-dihydroxyphenylacetaldehyde (DOPAL), an HT derivative, is implicated in intraneuronal dopamine metabolism. At low doses, DOPAL prevents α-synuclein-induced aggregation and neurotoxicity. However, at high concentrations, DOPAL can cause neurotoxicity and brain damage, accelerating PD pathogenesis [98]. Recent evidence suggests a functional relationship between the neuroprotective mechanisms exhibited by DOPAL involving the prevention and destabilization of α-synuclein fibril formation via the activation of stress resilience proteins such as Sirt-2, HO-1 and Hsp70 in a dose-dependent manner [100]. Human studies have investigated the effects of HT on cognitive function. A randomized study on 72 participants found that 3 g twice daily of desert olive tree pearls (DOTPs), containing 162 times more polyphenol HT than olive oil, enhanced cognition, specifically memory, attention, reaction time and executive function in middle-aged and older adults [101]. In addition, an interesting study by Han et al. revealed that dietary supplementation of HD (20 mg/kg) increased plasma testosterone, its derivative testosterone glucuronide, and also antioxidant molecules such as L-carnitine and its metabolite propionyl-L-carnitine, while decreasing bile acids and their derivatives. This regulates gut microbiota with a higher abundance of beneficial bacteria such as *Bifidobacterium*, *Lactobacillus*, *Eubacterium*, *Intestinimonas*, *Coprococcus*, and *Butyricicoccus*, and a lower abundance of harmful microbes such as *Streptococcus*, *Oscillibacter*, *Clostridium_sensu_stricto*, *Escherichia*, *Phascolarctobacterium*, and *Barnesiella*, thereby ultimately improving spermatogenesis and semen quality after 2 months. Therefore, HD could be used as dietary adjuvant to enhance semen quality [102]. Pharmacokinetic studies have shown that the bioavailability of HD-based food supplements increases in a dose-dependent manner. After HT-fortified olive oil intake, the main metabolites found in both plasma and urine are homovanillic acid, HT-3-O-sulphate, and 3,4-dihydroxyphenylacetic acid. The bioavailability of HT is a crucial prerequisite for its beneficial effects, as the maximum concentrations of HT in plasma peak 30 min after intake. This confirms that HT-based nutritional supplements may have good potential to counteract oxidative stress and inflammation in pathological conditions [103]. Finally, a double-blind, crossover, placebo-controlled trial demonstrated that polyphenols of melatonin and HT exerted antioxidant effects by reducing DNA damage and the production of F2-dihomo-isoprostanes, n-6 DPA and F4-neuroprostanes, and by protecting adrenic acid, docosahexaenoic and eicosapentaenoic acids from oxidative attack after red wine intake in healthy volunteers. Additionally, the intake of red wine that is rich in melatonin and homovanillic acid induces vasodilatory effects, probably mediated by the nitric oxide and increased plasma guanosine-3′-5′-cyclic monophosphate (cGMP) plasmatic levels, [75]. Overall, Hidrox^®^ and its potent polyphenolic compounds could represent novel nutritional candidates for preventing and treating CNS disorders related to oxidative damage, neuroinflammation, and steroid deregulation, thereby enhancing cognition and the quality of life in humans.

### 4.5. Camellia sinensis

Green tea, derived from the *Camellia sinensis* plant, is one of the most popular beverages worldwide. Its active polyphenols are thought to possess several biological properties, including chemoprevention, inhibition of tumor proliferation, steroid hormone regulation, antioxidant, and anti-inflammatory activities [104]. Dried leaves of this plant contain polyphenols (20–30%), primarily flavanols, known as catechins [105]. The predominant catechins include epigallocatechin-3-gallate (EGCG), epicatechin-3 gallate (ECG), epigallocatechin (EGC) and epicatechin (EC). Preclinical and clinical evidence has reported the protective effects of catechins on steroid hormone systems (Table 5) [104,106]. Notably, the major constituent in green tea extract, EGCG, blocks testosterone release in rat Leydig cells by significantly reducing the activity of the protein kinase A (PKA) and protein kinase C (PKC) signaling pathways and by directly or indirectly inhibiting both mitochondrial P450 side-chain cleavage enzyme (P450scc) and 17β-hydroxysteroid dehydrogenase (17β-HSD), which are essential for hormone synthesis. The study indicates that EGCG regulates 17β-HSD and leads to a reduction in cellular steroidogenic capacity [104]. Additionally, the major phenolic compound EGCG, present in aqueous tea extracts, decreases 11β-hydroxysteroid dehydrogenase type 1 (11β-HSD1) cortisol dehydrogenase activity in microsomes in a dose-dependent manner [107]. Importantly, intracerebroventricular injection (i.c.v.) of EGCG (50, 100 and 200 μg) attenuates stress behavior and plasma corticosterone concentration by inducing anti-anxiety, sedative and hypnotic effects via GABARs activation, in a dose-dependent manner [108]. Intriguingly, physiological/low concentrations of EGCG (5 μM and 10 μM) increased StAR expression and progesterone production by upregulating PKA/cAMP-responsive element-binding protein (PKA-CREB) signaling pathway in human granulosa cells [109]. Furthermore, high-dose EGCG supplementation (100 mg) during aging reinforced systemic immunity by enhancing cellular immune response through increased plasma DHEA levels, while a low dose of 25 mg of EGCG upregulated elements of the antioxidant defense system, such as SOD levels, to block oxidative stress and neuroinflammatory responses in aged albino rats [110]. In addition, low doses of EGCG (0.1 and 0.2 mM) in synergy with caffeine (10–30 mM) reduced total lipids, triglycerides and cholesterol in *C. elegans* [111]. A recent study by Siddiqui et al. demonstrated that EGCG, in synergy with vitamin D, strongly binds to human serum albumin to exert antiglycation ability by decreasing advanced glycation end products (AGEs), ROS overproduction-induced glycation levels, DNA damage and cytotoxicity [112]. Low oral administration of EGCG and/or ferulic acid (30 mg/kg each) once daily inhibited oxidative stress and neuroinflammatory markers (e.g., TNF-α and IL-1β), reverted cognitive impairment, and mitigated synaptotoxicity and cerebral vascular β-amyloid deposits in AD transgenic mice after 3 months [113]. Finally, a pilot study of 91 patients with uterine fibroids suggested that combined treatment with EGCG (300 mg), vitamin D (50 mg) and D-chiro-inositol (50 mg) in the form of 2 pills daily for 3 months reduced the time required to perform surgery and the bleeding during surgery without affecting liver function [114].

### 4.6. Vitamin D

Vitamin D is a neurosteroid hormone crucially involved in brain health and/or diseases [115]. Its endogenous concentrations influence multiple cellular pathways and mechanisms, including neural cell proliferation, neurotransmission, oxidative stress, inflammation and immune function in the CNS. Notably, vitamin D, when at adequate levels, acts as an important endogenous and/or exogenous regulator of neuroprotection, as it interacts with neurotransmitters and hormones to modulate various brain processes and pathways in animals and humans [116]. Emerging preclinical and clinical evidence shows a strong association between vitamin D deficiency and the onset of CNS disorders that impair brain development, leading to neuropathological conditions like AD, ASD, depression and schizophrenia, especially during pregnancy and early childhood [116,117,118,119]. Indeed, both insufficient (25–49.9 ng/mL) and deficient (<25 ng/mL) levels of vitamin D may contribute to increased AD susceptibility [116]. Equally important, high vitamin D deficiency (<20 ng/mL) is described in children with ASD and in pregnant mothers whose offspring will later develop ASD, suggesting a potential role of this neurosteroid hormone as a contributory risk factor in the etiopathogenesis of ASD [118,120]. Specifically, vitamin D deficiency caused by a polymorphism in the vitamin D receptor (VDR) gene is associated with lower levels of serum vitamin D in children with ASD, when compared with controls [121]. Interestingly, vitamin D supplementation at a dose of 100 μg/kg (three times a week for four weeks) provides brain health effects and improves neuronal synapse and memory, as well as abrogates amyloid beta (Aβ) production by upregulating the Nrf2 pathway and HO-1 and Sirt1 and by downregulating NF-κB pathway and pro-inflammatory cytokines, such as TNF-α and IL-1β in a rodent model of AD [122]. Furthermore, the neurosteroid vitamin D prevents and regulates nerve cell function to counteract the adverse effects of long-term glucocorticoid therapy with prednisolone (5 mg/kg b.w.) by upregulating 1a-hydroxylase (CYP27B1) expression, an important component of the vitamin D–auto/paracrine system and inhibiting the NF-κB pathway in rats [123]. Recent findings in humans, specifically in 56 patients with mild to moderate depression, randomly assigned to receive vitamin D (50,000 IU 2 wks^−1^) or placebo for eight weeks, showed that vitamin D intake significantly increased serum 25(OH)D concentrations, accompanied by improvements in symptom severity, independently of circulating IL-1β, IL-6, and hs-CRP concentrations (Table 6) [124]. Overall, the data indicate that vitamin D deficiency is a critical risk factor for the development of neurological disorders; therefore, its proper intake through dietary interventions and/or supplements may promote brain health in humans.

### 4.7. Omega-3 Fatty Acids

The brain is particularly rich in polyunsaturated fatty acids (PUFAs), including omega-6 (ω6) and omega-3 (ω3) fatty acids. Recently, PUFAs have gained considerable attention for their potential role in the prevention and therapy of neurodegenerative and psychiatric diseases, such as AD, PD, ASD, anxiety and depression [125,126,127]. PUFAs include arachidonic acid (AA), docosapentaenoic acid (DPA), docosahexaenoic acid (DHA) and eicosapentaenoic acid (EPA). Interestingly, neurosteroids regulate the synthesis of PUFAs in brain cells [126]. Notably, 17β-estradiol has been shown to influence the production of EPA, DPA and DHA from alpha-linolenic acid (ALA) in neuroblastoma SH-SY5Y cells via the activation of the peroxisome proliferator-activated receptors (PPAR)-signaling pathway [126]. A pilot study has reported that DHA deficiency is closely linked with increased corticotropin-releasing hormone, contributing to HPA axis hyperactivity. Conversely, elevated levels of neuroactive steroids, allopregnanolone and 3α,5α-tetrahydrodeoxycorticosterone (THDOC) appear to counter-regulate HPA hyperactivity under pro-oxidant conditions [127]. This study has indicated that psychiatric patients with alcoholism and depression exhibited lower plasma DHA levels and higher plasma concentrations of the neuroactive steroid DHP, whereas healthy subjects showed lower plasma DHA levels and higher plasma THDOC and isopregnanolone (3beta, 5alpha-THP) levels, likely contributing to increased feedback inhibition of the HPA axis [127]. Moreover, a cross-sectional study by Semba et al. has demonstrated that children exposed to tobacco smoke had perturbations in metabolic pathways, including lower serum ω-3 and ω-6 long-chain PUFA levels, which are fundamental for growth and development; lower sulfated neurosteroids levels, which are involved in brain development; lower carnitine levels, which are essential for β-oxidation of fatty acids; altered glutathione metabolism; and higher serum concentrations of cotinine, a marker of tobacco exposure, resulting in childhood stunting [128]. Additionally, a randomized trial has observed that supplementation with EPA and DHA reduces thromboxane B (TXB) levels and increases BDNF and vitamin D levels in children and adolescents with depressive disorder after 12 weeks [129]. More recently, the same authors have reported that ω-3 fatty acids stimulate both the kynurenine/tryptophan (KYN/TRP) ratio and the biopterin and serotonin (5-hydroxytryptophan (SER)) pathways, whereas ω-6 fatty acids increased only the KYN/TRP ratio in the urine of depressed children and adolescents after 12 weeks (Table 7) [130]. Finally, a case report study has suggested that ω-3 therapy and co-supplementation with vitamin D may be potentially effective in treating the core symptoms of ASD [131].

## 5. Polyphenol–Nanoparticle Delivery Systems and Neurosteroid Signaling

Nowadays, nanomedicine platforms increasingly represent an innovative approach for drug delivery in the brain. Polyphenol–nanoparticle delivery systems improve the bioavailability and overall stability of circulating polyphenols or neurosteroid hormones, which then diffuse more efficiently across the BBB, promoting greater brain resilience and health [132].

### 5.1. Preclinical Studies 

Growing evidence has shown that progesterone, when encapsulated within hydrophobically modified hyperbranched polyglycerol, increases its solubility, stability, and bioavailability in both cells and animal models [133]. Importantly, Aneesh et al. have synthesized a photoactivatable dextran–DHEA conjugate and encapsulated it within DNA icosahedra bearing lipid motifs, which released the neurosteroid DHEA. This ultimately provided neuronal activation and survival by modulating crucial processes such as NMDA-induced excitotoxicity and neurogenesis in vitro and in *C. elegans* [134]. Likewise, functional nutrients, including RSV at a low dose of 20 mg orally once daily, exerted potent antioxidant capacity and upregulated steroidogenesis-related gene expression (3 β-HSD, 17 β-HSD, and Nr5A1) by restoring the significantly iron oxide (Fe_2_O_3_) nanoparticle-induced depletion of testosterone, follicle-stimulating hormone, luteinizing hormone, and testicular antioxidant enzymes [135]. Furthermore, RSV coated with gold nanoparticles prevents 17β-estradiol/ERα-induced neuroglobin accumulation and induces apoptosis in cancer cells (Table 8) [136]. Interestingly, treatment with zinc oxide–RSV nanoparticles attenuate the harmful side effects of levofloxacin. Specifically, zinc oxide–RSV nanoparticles significantly augment testicular antioxidant SOD activity and reduce MDA levels in rats treated with levofloxacin. Therefore, zinc oxide–RSV nanoparticles could be a possible solution for levofloxacin-induced fertility problems [137]. Consistent with brain health, recent evidence has shown that intravenous injection of CUR loaded with T807-modified nanoparticles at a low dosage of 5 mg/kg effectively cross the BBB, improving its permeation into the brain. In this way, CUR loaded with T807-modified nanoparticles shows stronger binding to the hyperphosphorylated form of tau protein in neurons, thereby decreasing its levels and blocking neuronal apoptosis and AD progression both in vitro and in vivo [138]. Intriguingly, it has also been shown that CUR-loaded lipid–core nanocapsules, at a low dose of 10 or 1 mg/kg p.o., provided higher neuroprotection than a high dose of 50 mg/kg of free CUR by reducing Aβ1-42-induced inflammatory cytokines (e.g., TNF-α, IL-6, IL-1β and IFN-γ) levels in serum and in the prefrontal cortex and hippocampus of aged mice [139].

### 5.2. Clinical Studies 

A randomized controlled study on 40 subjects investigated the effects of the highly bioavailable daily oral form of CUR, Theracurmin^®^, on memory performance. The study demonstrated that Theracurmin^®^ containing 90 mg of CUR, when taken twice daily, enhances memory and attention in middle-aged and non-demented adults. These brain health effects were found to be strongly linked to reductions in amyloid plaque aggregates and tau accumulation in the hypothalamus and amygdala after 18 months of treatment without any toxic effects [140]. Finally, a more recent double-blinded, placebo-controlled parallel-group comparative study found that supplementation of CurQfen^®^, another highly bioavailable form of CUR, provided significant BBB permeability and brain bioavailability. Specifically, a dose of 400 mg × 2/day of CurQfen^®^ attenuates AD progression and improves locomotor and cognitive functions by upregulating BDNF levels and downregulating IL-6 and TNF-α cytokines in patients with moderate dementia after 6 months [141].

## 6. Neurosteroids as Modulators of Neuroinflammation

Neuroinflammation is a physiological adaptive mechanism involving the activation of immune cells and the release of neuroinflammatory mediators in response to infection and/or injury in CNS. Notably, chronic or sustained activity and proliferation of glial cells (astrocytes) can trigger neurotoxicity, increase BBB permeability and contribute to the onset of neuropathological mechanisms, including neuronal death, changes in gene expression, neurotransmitters, and altered cognitive function, ultimately exacerbating the progression of various brain lesions and disorders [142]. Of note, both excitatory neurons and glia synthesize neurosteroids that are substantially affected by neuroinflammation and oxidative stress [143]. However, neurosteroids such as 17β-estradiol, DHEA, progesterone and allopregnanolone, produced de novo in the CNS or derived from the circulation, can regulate neuroinflammation and support neuronal survival during degenerative and neuropsychiatric disorders [144]. In line with neurohormesis, endogenous neurosteroids are emerging as innovative therapies for CNS disorders due to their potent anti-neuroinflammatory and neuroprotective effects in a dose-dependent manner [145,146,147,148,149,150,151,152,153,154].

### 6.1. In Vitro Studies

Recent preclinical evidence reports that the neurosteroid allopregnanolone (3α,5α)3-hydroxypregnan-20-one, 3α,5α-THP) inhibits pro-inflammatory factors, including monocyte chemoattractant protein-1 (MCP-1), high mobility group box 1 (HMGB1), and tumor necrosis factor alpha (TNF-α) activated by MyD88-dependent Toll-like receptor (TLR) pathways and enhances the anti-inflammatory IL-10 and BDNF levels in both macrophage cells and rat brain [145,146]. In addition, pregnenolone, 5α-dihydroprogesterone (5α-DHP) and pregnanolone modulate neuroinflammation in response to the neurotoxic insults induced by rotenone during neurodegenerative diseases in microglial BV-2 cells [147]. Interestingly, allopregnanolone, pregnanolone and 3α,5α-THDOC exhibit a remarkable ability to regulate primary mediators of inhibitory neurotransmitter systems in the brain, especially GABARs. These neurosteroids bind to specific sites on the GABARs in both microglia and astrocytes and mediate anti-inflammatory effects through inhibition of the NF-kB pathway [148]. Furthermore, a synthetic neuroactive steroid, C-17-spiro-cyclopropyl DHEA derivative (ENT-A010), has been shown to be a neuroprotective and pro-resolving factor in inflammatory states, promoting neuronal survival and enhancing microglial phagocytic capacity to inhibit the neuroinflammatory process in both mouse hippocampus and microglial cells via activation of tropomyosin-related kinase A (TRKA) receptor signaling [149]. Moreover, BNN27, a prototype based on dehydroepiandrosterone, mimicking neurotrophin NGF, provided neuronal survival in PC12 cells through the activation of the TRKA receptor against neurodegenerative process [150]. In addition, treatment with exogenous allopregnanolone (1 nM) efficiently prevents the reduction of cell viability in human microglial cells after oxidative damage induced by rotenone exposure [151].

### 6.2. In Vivo Studies

A recent study investigated two novel synthetic water-soluble neurosteroids, valaxanolone and lysaxanolone, as potent neuroprotectants in combating the long-term behavioral and neuropathological impairments caused by acute organophosphate intoxication leading to status epilepticus. Valaxanolone and lysaxanolone, at a low dose of 10 mg/kg, markedly reduced anxiety and memory deficits as well as aggressive and depressive-like behaviors; and prevented the di-isopropylfluorophosphate-induced chronic loss of principal neurons and GABAergic inhibitory interneurons in the hippocampus; decreased the inflammatory response (e.g., astrogliosis and microgliosis) in animal models of neurodegeneration, particularly in the hippocampus and amygdala [152]. Similarly, low doses of ganaxolone (10 mg/kg) effectively mitigated long-term neurodevelopmental disability, cognitive impairment and neuroinflammation by reducing GFAP (+) astroglial and IBA1(+) microglial expressions in the hippocampus and amygdala of a pediatric rat model of acute diisopropyl-fluorophosphate exposure [153]. Recent findings have demonstrated the neuroprotective effects of both progesterone and allopregnanolone in the mouse model of motoneuron degeneration by preserving mitochondrial respiratory complex I activity via the induction of the antioxidant enzyme SOD [154]. Neuroinflammation enhances GABAergic neurotransmission in the cerebellum by upregulating the TNFR1-glutaminase-GAT3 and TNFR1-CCL2-TrkB-KCC2 pathways. In light of this, a study conducted by Mincheva et al. has observed that a low dose of golexanolone, 50 mg/kg/day orally administered, reduced TNFα and astrocyte activation and increased the anti-inflammatory IL-10 in plasma and the cerebellum by attenuating the neurotransmission of GABARs and improved cognitive and motor function in hyperammonemic rats [155]. Furthermore, a low dose of exogenous allopregnanolone (1 μM) prevented LPS-induced learning defects on long-term potentiation (LTP) through an inhibition of TLR4-indendent mechanism in the hippocampus in vivo [156]. Pregnenolone is believed to be a vital precursor to all neurosteroid hormones and is associated with several metabolic pathways, including the synthesis of corticosteroids (cortisol and cortisone), neuroactive steroids (progesterone) and their metabolism into their reduced derivatives, such as 5α-DHP, allopregnanolone and 11-deoxycorticosterone [157]. In particular, human microglia produce and release neurosteroids and this metabolic activity could be modulated under neurotoxic conditions. Accordingly, a recent in vivo study has shown sex-specific metabolic signatures in the hypothalamus in response to LPS exposure [158]. The study observed that the levels of pregnenolone and progesterone decreased, whereas 5α-DHP, allopregnanolone and 11-deoxycorticosterone levels increased in LPS female mice. These data indicate that steroid biosynthesis in the hypothalamus of female mice is perturbed under inflammatory stress and that pregnenolone supplementation or inhibition of hypothalamic 5α-reductase type I restores neuronal homeostasis and mitigates excessive neuroinflammatory response [158]. Overall, innovative preventive and pharmacological interventions using neurosteroids alone and/or in synergy with functional nutrients, such as curcumin, resveratrol and polyphenol nanoparticles targeting antioxidant pathways and anti-neuroinflammatory signaling, could represent promising therapeutic candidates for future clinical trials in order to prevent or inhibit neuroinflammatory cascade leading to BBB permeability and dysfunction and the onset of severe nervous system disorders.

## 7. The Role of Neurosteroids in Nervous System Disorders: Focus on Nutrients

Neurosteroids are dose-dependent, potent modulators of neurotransmitters, synaptic plasticity, neuronal function and neurogenesis in the CNS [159]. In the brain, aberrant levels of the neurosteroids DHEA, DHEAS and pregnenolone that occur during aging and acute stress play a key role in the development and progression of AD, PD, autism and depression [160]. Adequate levels of circulating serum DHEA have been considered an important marker of human longevity and lifespan extension [161]. Emerging evidence has revealed that functional food nutrients provide brain health effects and may interact with neurosteroids to restore neuronal redox homeostasis during neuropathological conditions (Figure 2) [58,123,162,163].

### 7.1. Alzheimer’s Disease

Alzheimer’s disease (AD) is a multifactorial age-related neurodegenerative disorder that currently has no effective therapeutic interventions to prevent or slow its progression.

Growing evidence shows that neurosteroids exhibit pleiotropic neuroprotective effects during AD pathogenesis [164,165,166,167]. Accordingly, physiological levels of neurosteroids act as neuroprotectors, targeting several crucial pathways and mechanisms involved in neuronal apoptosis, oxidative stress, neuroinflammation, mitochondrial dysfunction and synaptic loss. During aging, alterations in homocysteine and DHEAS levels are closely related to GSH depletion and increased oxidative damage leading to the development of AD [164].

#### 7.1.1. Preclinical Studies 

Interestingly, Calan et al. have reported that high neurotoxic Aβ concentrations enhanced pregnenolone levels likely through a cellular self-defense mechanism in a dose and time-dependent manner in neuronal cells [165]. In addition, recent research has examined changes in the expression of neurogenesis genes under the influence of LPS on neurosteroidogenesis in human neuroblastoma SH-SY5Y cells. The authors detected that SH-SY5Y cells increased β-amyloid deposition and apoptosis, leading to depletion of DHT and DHP concentrations [166]. Furthermore, physiological/moderate concentrations of DHEAS activate the expression of tight junction (TJs) proteins such as zonula occludens-1 (ZO-1) and claudin-3 in endothelial cells. In addition, progesterone is significantly decreased and 17β-estradiol increased in the prefrontal cortex and hippocampus of Aβ-treated rats as well as in SH-SY5Y cells [167,168]. Subcutaneous injection of progesterone has been found to reverse Aβ-mediated neuroinflammation and upregulation of TNF-α and IL-1β in a dose-dependent manner in vivo [168]. Conversely, allopregnanolone content remained unchanged in both the hippocampus and prefrontal cortex of Aβ-treated rats [168]. This shows that Aβ_25–35_ targets only specific steroidogenic enzymes, specifically reducing the activity of 3β-HSD and progesterone without affecting the 3α-HSD involved in allopregnanolone synthesis. Thus, DHEAS is directly involved in the formation and maintenance of BBB integrity for the prevention or treatment of neurological disorders [164]. Importantly, GSH is a cofactor for hydroxysteroid sulfotransferase, one that converts DHEA to its sulphated derivative, DHEAS. Therefore, reduced GSH levels in the aged brain may lead to lower DHEAS content in the latter. Consistent with this observation, a study has demonstrated that dietary supplementation with N-acetyl cysteine (50 mg), α-tocopherol (3 mg) and α-lipoic acid (1.5 mg) has the capacity to remove oxidative burden and restore glutathione content in the brain, thereby increasing homocysteine and DHEAS levels with promising therapeutic effects in AD [169]. Similar to polyphenols, pregnenolone and allopregnanolone exerts neuroprotection by blocking and attenuating Aβ25–35-induced loss of memory and lipid peroxidation in AD and in dentate gyrus mouse models via activation of sigma (σ_1_) receptor-dependent PI3K-Akt-mTOR signaling in a concentration-dependent manner [170,171].

#### 7.1.2. Clinical Studies

Human studies demonstrated that endogenous neurosteroids, such as DHEA, and pregnenolone levels are elevated in the temporal cortex of AD patients [172], whereas allopregnanolone is reduced in the same brain tissue [173]. To note, DHEAS has shown anti-glucocorticoid activity as it is capable of blocking elevated cortisol levels related to stress on anxiety and depression, demonstrating positive effects on cognitive decline in patients with dementia [174]. Clinical studies have supported the notion that supplementation with moderate/low doses of DHEA in older women with cognitive decline and lower DHEA levels is effective in treating cognitive disorders. Notably, twelve women who received DHEA capsules (25 mg/day) for six months showed improvements in neuronal function and maintenance of basic activities of daily living [175].

### 7.2. Parkinson’s Disease

Parkinson’s disease (PD) is the second most widespread neurodegenerative disorder characterized by a massive loss of dopaminergic cells in the substantia nigra, leading to dopamine hypofunction and alteration of the basal ganglia circuitry. Neuroactive steroids are modulators of neurotransmitter systems and may thus help to control PD symptoms and the adverse effects of dopaminergic drugs [176]. The substantia nigra of the human brain expresses high concentrations of allopregnanolone, which positively modulates the action of GABA at GABARs, and especially of 5α-DHP, a neurosteroid that acts at the genomic level on the dopaminergic system, which is extremely compromised in PD [176].

#### 7.2.1. Preclinical Studies 

Recently, Castelnovo et al. has demonstrated that both the physiologic agonist progesterone and the specific membrane progesterone receptor agonist Org OD 02-0 are effective in decreasing the neuronal cell death induced by 6-hydroxydopamine (6-OHDA) and 1-methyl-4-phenylpyridinium (MPP^+^) via activation of the ERK and PI3K-AKT signaling pathways [177]. Preclinical studies have shown that progesterone protects dopaminergic neurons from degeneration [178]. Preservation of striatal dopamine by progesterone in the brain of a mouse model of PD has been demonstrated [178]. In line with this notion, progesterone increases dopamine release and gene expression of dopamine transporter in the striatum [179]. A recent study has highlighted that pregnenolone, even at a low dose of 6 mg/kg, effectively prevents L-DOPA-induced dyskinesias by decreasing 6-OHDA lesions in the rat model of PD without affecting L-DOPA-induced motor improvements and its ability to modulate striatal BDNF levels [160].

#### 7.2.2. Clinical Studies 

Importantly, a positive interrelationship between patients with PD and decreased vitamin D levels associated with cognitive impairment has been documented [180]. Gas chromatography–mass spectrometry analysis revealed that progesterone metabolites, such as 5α-DHP and allopregnanolone, significantly increase in the early stages of PD in the substantia nigra and prefrontal cortex tissues to induce neuroprotection. On the other hand, these metabolites are downregulated in the advanced stages of PD, exacerbating its progression [181]. Other evidence shows the preventive and therapeutic potential of neurosteroids in animal models of PD, though their clinical efficacy has not yet been clarified [182]. Taken together, innovative therapeutic interventions with low doses of 5α-DHP, progesterone and allopregnanolone, alone or in synergistic treatment with functional food nutrients that stimulate the synthesis of endogenous neuroactive steroids and positively regulate GABA through GABARs, could provide a neuroprotective approach with clinical potential in healthy subjects with genetic PD predisposition and in particular when their production is impaired, as occurs in advanced stages of PD.

### 7.3. Depression 

Depression is a neuropsychiatric disorder that severely limits psychosocial functioning and impairs quality of life. Preclinical and clinical evidence has highlighted that dysregulation of neurosteroid production plays a prominent role in the development of depressive disorders, especially in women [183].

#### 7.3.1. Preclinical Studies 

Equally important, functional food nutrients including phenols, flavonoids and probiotics, but also vitamin D, have received considerable attention in terms of the prevention and therapy of major depressive disorders and in maintaining or restoring steroid hormone homeostasis in the brain [184,185]. Catalpol in particular, a most abundant iridoid glycoside that is present in *Radix Rehmannia glutinosa Libosch*, has shown antidepressant effects by attenuating corticosterone-induced depressive-like behaviors in mice via inhibition of HPA axis hyperactivity, central inflammation and oxidative damage through dual modulation of NF-κB and Nrf2 pathways [184]. In addition, it has been determined that *Lactuca serriola* dried seed extracts and their active flavonoids, particularly quercetin, phenols and terpenoids, at an oral dose of 300 mg/kg, induced significant anti-seizure activity and anxiolytic effects by upregulating SOD, CAT and total GSH via GABARs activation in the brain [185]. Finally, treatment with the probiotic Komagataella pastoris KM71H (8 log UFC·g^−1^/animal, intragastric route) prevented restriction stress-induced depression-like behaviors and LPS challenges by regulating BBB permeability, attenuating proinflammatory NF-κB, IL-1β and IFγ cytokines and indoleamine 2,3-dioxygenase levels in the hippocampus, prefrontal cortex and intestine, mediated by the depletion of plasma corticosterone levels, in mice [186].

#### 7.3.2. Clinical Studies 

Importantly, low levels of allopregnanolone have been correlated with an increased risk of depression during late pregnancy [187]. This occurs because serum allopregnanolone levels range from 0.5 to 5 nmol/L with the stages of the menstrual cycle. However, serum allopregnanolone concentration rises markedly, up to more than 10-fold during pregnancy, and then rapidly decreases to 1–2 nmol/L after the early postpartum period, suggesting a potential mechanism of postnatal depression [188]. A recent randomized clinical study showed that transdermal estradiol (0.1 mg/d) plus intermittent micronized progesterone (200 mg/d for 12 days) treatment for twelve months was more effective than placebo in preventing the development of clinically significant depressive symptoms among initially euthymic peri-menopausal and early post-menopausal women [189]. Furthermore, a preliminary clinical study in women with anxiety and depressive peripartum and postpartum symptoms has suggested that higher levels of allopregnanolone reflect a perturbation in its targets, particularly reduced sensitivity or plasticity of GABARs [190]. A randomized study revealed that treatment with brexanolone 90 μg/kg/h or placebo was associated with rapid improvement in depressive and anxiety symptoms and insomnia compared with placebo in women with postpartum depression [191].

### 7.4. Autism Spectrum Disorder

Autism spectrum disorder (ASD) is a multifactorial neurodevelopment disorder influenced by a complex interaction between genetic, immune and environmental factors, leading to the development of altered cortical circuits, atypical trajectories of brain maturation and behavior, impaired neurogenesis, synaptogenesis and imbalance of excitatory and inhibitor neurotransmitter systems, which ultimately contribute to alterations in social, repetitive behavioral and intellectual disability. Recently, preclinical and clinical evidence has focused on the strong association between deregulation of neurosteroids, particularly pregnenolone and allopregnanolone and ASD symptom severity [6,192,193].

#### 7.4.1. Preclinical Studies 

Placental allopregnanolone deficiency has been shown to alter cortical GABAergic signaling, leading to the development of ASD symptoms in both mice and humans [192]. Furthermore, a moderate dose of ganaxolone (20 mg/kg) improved sociability and autism-related repetitive behaviors in mice through the activation of GABARs [193]. Interestingly, subcutaneous administration of ganaxolone, a β-methylated analogue of allopregnanolone, in preterm guinea pigs protected against the loss of myelination, hyperactive behavior and the premature mortality observed in untreated preterm control animals, highlighting the neuroprotective potential of allopregnanolone in mitigating neurodevelopmental disorders associated with preterm births [194].

#### 7.4.2. Clinical Studies 

A recent clinical study performed by Chew et al. demonstrated that lower serum allopregnanolone levels correlated with more severe restricted and repetitive behaviors in adult males with ASD [195]. Other research indicates that fetal immune state closely resembles that of the mother [196] and neuroactive steroids are transferred from the mother’s placenta to the fetus and are also synthesized within the fetal organism [196]. This phenomenon explains the increase in progesterone and allopregnanolone levels after birth compared with the fetal phase [197]. Indeed, poor myelination, low birth weight, increased mortality and encephalopathy are closely associated with higher endogenous concentrations of allopregnanolone, progesterone, IL-6, and IL-10 among preterm newborns [198]. In addition, maternal PCOS and hirsutism are associated with an increased likelihood of autism [199,200]. More recently, a clinical study investigated the effects of testosterone, 17-hydroxyprogesterone (17-OHP), and cortisol in serum collected during the first trimester from mothers of children with autism and mothers of unaffected population-based control children. The study demonstrated that altered serum maternal steroid hormonal levels in early pregnancy may promote the onset of autism. Notably, moderate/low levels of maternal serum estradiol are strongly implicated with intellectual disability, whereas high concentrations of cortisol and 17-OHP are associated with autism without intellectual disability in the offspring [201]. It is noteworthy that neurosteroids can directly act on GABA, which is responsible for most of the fast-synaptic inhibition in the brain [202]. Pregnenolone, in particular, is a neurosteroid with modulatory effects on GABA neurotransmission [203]. A recent clinical study on 59 patients with ASD who were randomly allocated to receive either pregnenolone or a matching placebo plus risperidone showed that pregnenolone, in synergy with risperidone, attenuated core ASD features [203]. The authors also report that pregnenolone adjunct to risperidone attenuated irritability, hyperactivity and stereotypy behaviors by regulating the GABAergic system in adolescent patients with ASD [203]. Additionally, genetic variants in the GABAR subunit genes *GABRB3*, *GABRG3*, and *GABRA5*, located on chromosome 15q11-q13 have been strongly associated with the onset and progression of ASD [204]. Overall, the data highlight that deregulation of circulating neurosteroids in the brain, particularly pregnenolone and allopregnanolone, may contribute to ASD symptomatology and severity. Currently, the impact of neurosteroids on ASD remains elusive. Future investigations in this promising field concerning neurosteroid signaling in health and/or disease could better elucidate the potential molecular mechanisms underlying their neuroprotective action, especially in synergistic combination with functional food nutrients, in order to develop innovative nutritional supplements for the prevention and management of neurodevelopmental disorders.

## 8. The Interaction of Food Nutrients and Neurosteroids via GABARs

Food nutrients and neurosteroids modulate endogenous GABAergic neurotransmission via GABARs [205,206,207]. The interest in polyphenols derived from medicinal plants is associated with the need to find new active chemicals different from classical drugs, avoiding several undesirable effects, such as tolerance, abstinence, dependence and memory disorders. Recent literature suggests the action of some phenols as positive modulators of GABARs, which encourages their use in the development of new natural drugs with therapeutic potential (Figure 3) [205,206]. Interestingly, phenols cross the BBB and can interact with neurosteroids, exerting health-promoting effects in the brain [72,208].

### 8.1. Flavonoids

Simple phenols such as catechol, resorcinol, pyrogallol and phloroglucinol, as well as complex polyphenols, including flavonoids, stilbenes, lignans, terpenoids and polyacetylenic alcohols, can, due to chemical and structural variability, selectively interact with different binding sites on GABARs, allowing them to be activators, blockers or allosteric ligands of GABA activity by inducing negative or positive effects on GABARs in the CNS in a dose–response mode [209,210,211]. Preclinical evidence has shown that phenols display hormetic actions on the Cl^−^-ATPase activity of β3-containing GABARs isolated from rat and fish brains with receptor stimulation at low doses and inhibition at high concentrations, thus following a biphasic dose–response curve [209,210]. Several earlier studies have reported that some isoflavones, such as genistein and their inactive analogue daidzein or tyrphostin at a dose of 100 µM, behave as protein tyrosine kinase (PTK) inhibitors to directly block GABAR function by a concentration–response curve, completely independent of any inhibition of endogenous tyrosine kinases [211]. Importantly, flavonoids and terpenoids, being lipophilic, easily cross the BBB and target GABARs to influence brain function and neurosteroid signaling; therefore, these are also called “a new family of benzodiazepines” as benzodiazepines reduce oxidative stress via GABARs [212,213]. A study reports that micromolar dose (30 μM) of prenylflavonoids anthohumol (XN), isoxanthohumol (IXN) and 8-prenylnaringenin (8PN) positively modulated GABA-induced responses in native and αβγ/δ recombinants [214]. In addition, low doses of 30 μM of rosmarinic acid activated GABARs through the interaction between its amino acid residues Arg192, Arg196, and Ser209 in cerebro-cortical synaptosomes to decrease Ca^2+^ influx and CaMKII/synapsin I pathway and inhibit the evoked glutamate release [215]. RSV (500 μM) can directly act on substantia gelatinosa neurons by the activation of GABARs and/or glycine receptors in a dose-dependent manner [216]. Interestingly, subchronic administration of RSV (50 mg/kg) alone or in combination with rufinamide (50 mg/kg) induced notable anxiolytic-like effects by inhibiting GABA reuptake transporter 1 protein, leading to increased synaptic levels of GABA neurotransmitter in rodents [217].

#### 8.1.1. Quercetin, Apigenin and Genistein 

Of note, the flavonoids of *Parkia roxburghii* (family Mimosaceae), including quercetin, catechin and biochanin A, alone and/or in combination, exhibited inhibitory effects on scopolamine-induced memory impairments in mice. This mechanism assumes that flavonoids have the capacity to reduce inflammatory markers (IL-6 and TGF-β) and inducible nitric oxide synthase (iNOS) and inhibit acetylcholinesterase (AChE), γ-aminobutyric acid A receptor, alpha5 (GABA_A_α5), glycogen synthase kinase-3 (GSK-3β), p38 mitogen-activated protein kinase (p38α/MAP-kinase), signal-regulated kinase and protein-serine/threonine kinase (ERK/Akt) and increase antioxidant enzymes (SOD, GSH and glutathione reductase) in the brain [218]. Moreover, apigenin and genistein act as GABA antagonists at flumazenil-insensitive α1β2 GABARs [219]. In addition, apigenin (1 mM) and EGCG (0.1 mM), at lower concentrations, markedly enhanced GABA responses by diazepam on the activation of recombinant GABARs in the presence of allopregnanolone, whereas higher doses inhibited these responses [219]. Furthermore, a recent study performed by Bappi et al. has revealed that quercetin (50 mg/kg, p.o.) and sclareol (1, 5, and 10 mg/kg, p.o.) increased the latency and decreased sleeping time against thiopental sodium-induced sleep in mice by binding GABARs, especially the α2, α3, and α5 subunits [220].

#### 8.1.2. Chrysin

Systemic injections of flavonoid chrysin (5,7-dihydroxyflavone) at ranging doses of 0.25, 0.5, and 1 μg exerted anxiolytic- and anti-despair-like effects in ovariectomised and cycling female rats via activation of GABARs. In proestrus, 0.5 and 1 μg of chrysin and allopregnanolone promoted anxiogenic-like behaviors. In dioestrus, chrysin and allopregnanolone (0.5 μg) provided anxiolytic-like effects. Picrotoxin, bicuculline and flumazenil prevented the effects of chrysin and allopregnanolone in both proestrus and dioestrus. Therefore, similar to the actions of allopregnanolone, the effects of chrysin in regulating anxiety-like behaviors are mediated by GABAR interaction in the dorsal hippocampus [221]. Overall, the above findings confirm that flavonoids act as GABAergic agents, similar to neurosteroids, as they can determine neurobiological changes ranging from negative action (e.g., simple phenols) to positive effects (e.g., polyphenols) on receptor activity in CNS disorders.

### 8.2. Tannins 

Emerging evidence has begun to highlight the promising therapeutic potential of tannins in modulating GABARs. Interestingly, phlorotannin supplement, a natural polyphenol derived from brown algae, and especially its component dieckol, acts as a positive allosteric activator of GABA_A_-BZD receptors and inhibits synaptic networks by increasing GABAergic transmission in a dose-dependent manner [222]. Therefore, dieckol could be considered a natural candidate for the treatment of psychiatric diseases such as anxiety and insomnia [222]. In addition, punicalagin (100 and 300 mg/kg), an ellagitannin contained in *Punica granatum* L., attenuates the effects of vincristine (75 µg/kg i.p.)-induced neuropathic pain by activating GABARs and inhibiting pro-inflammatory cytokines (e.g., TNF-α and IL-6) in silico and in vivo [223].

### 8.3. Terpenoids 

Terpenoids are a group of natural compounds composed by the condensation of isoprene units with the potential to modulate GABARs in the CNS. Increasing evidence has demonstrated that sesquiterpenoids from *Valeriana officinalis* L., are used to treat anxiety and sleep disorders by binding GABARs on the β isotype [224]. Moreover, bilobalide a sesquiterpene lactone, present in Gingko biloba, exhibits several actions, including anticonvulsant and antagonistic action at ⍺1β2γ2L GABARs, similar to bicuculline and picrotoxin, via a presynaptic route to maintain GABA levels and hence glutamate decarboxylase (GAD) activity in the hippocampus and cerebral cortex [225]. Monocyclic and bicyclic sesquiterpenes, including ⍺-humulene and β-caryophyllene, and sesquiterpenoids, such as guaiol, are the major components of the volatile fraction of hop but are also part of many other essential oils. Importantly, a low dose of 600 μM of ⍺-humulene, β-caryophyllene and guaiol exhibited a negative allosteric modulation by binding specific γ2 and δ subunits of GABARs that reduced the GABAergic neurotransmission in neurons via interactions of the endogenous ligand pregnanolone to the receptor [58]. In addition, curcumol, a sesquiterpene compound and a major bioactive component of Rhizoma Curcumae oil, at a concentration of 50 μM, behaves as a positive allosteric enhancer of GABARs in hippocampal neurons, but acts at a different site than benzodiazepines [226]. Intriguingly, a meroterpenoid, chrodrimanin B, produced by *Talaromyces* sp., has been shown to act as a competitive antagonist of the silkworm larval orthosteric site of GABARs in the rat brain [227]. In addition, artemisinins, a group of plant-derived sesquiterpene lactones, act as effective antimalarial agents by modulating GABARs. Accordingly, a recent study has demonstrated that artemisinins bind to gephyrin, the multifunctional scaffold of GABAergic synapses and modulate inhibitory neurotransmission in vitro and in pre-symptomatic AD mice [228]. Notably, a low dose of artemisinins (10 mg/kg) exerted neuroprotective action in the hippocampus of mice by increasing the level of gephyrin, gephyrin phosphorylation at Ser270 and GABAAR-γ2 and cyclin-dependent kinase 5 (CDK5)/p35 signaling, potentially affecting GABAergic inhibitory synapses [228].

### 8.4. Neurosteroids 

The neuroprotective mechanisms of neurosteroid actions are principally triggered by binding to GABARs as agonists to regulate neural functions [229]. They can be positive or negative regulators of GABAR function, depending on the chemical structure of the steroid molecule [229].

#### 8.4.1. Allopregnenolone, THDOC and Androstanediol

Brain allopregnanolone, THDOC and androstanediol are positive allosteric modulators of GABARs and the primary neurosteroid inhibitors that elicit sedative, anxiolytic and anticonvulsant actions. Specifically, the binding site on GABARs is located between α_1_ and β_2_ subunits [207,229]. However, neurosteroids bind multiple subunits of GABARs, displaying a preferential affinity at extra-synaptic δ-containing receptors and subsequent induction of tonic inhibition during epilepsy and postpartum depression in women [230].

#### 8.4.2. Pregnenolone Sulfate, DHEA and DHEAS

Sulfated neurosteroids, such as pregnenolone sulfate and DHEAS, are negative GABAR modulators and act as memory-enhancing agents in a dose-dependent manner [229]. Nevertheless, sulfated neurosteroids are potent allosteric agonists of NMDA and σ_1_ receptors [229]. In the brain, oxidative stress induces changes in GABARs due to the alterations of endogenous neurosteroids. Under stress conditions and the increased cellular ROS caused by neural injury such as ischemia, trauma or neurodegeneration, glia cells, particularly oligodendrocytes and astrocytes but not neurons, may act as reservoirs of DHEA by using the alternative signaling pathway induced by Fe^2+^. Indeed, it has been shown that treating oligodendrocytes with β-amyloid enhanced intracellular free radicals and DHEA production in vitro [231]. Therefore, DHEA synthesis can be regulated by intracellular free radicals and treatment with vitamin E is capable of blocking β-amyloid and ROS production [231]. Interestingly, other researchers have observed that it is mainly astrocytes and only partially neurons that have the ability to synthetize and metabolize DHEA to testosterone and estradiol in a dose-dependent manner via the cytochrome P450 17alpha-hydroxylase pathway [232]. Moreover, the anti-stress hormone DHEAS provides beneficial effects to hypoxic processes by increasing survival and improving stress resistance, ultimately attenuating injuries to the GABAergic system induced after sodium sulfite exposure in *C. elegans* [233].

#### 8.4.3. Pregnanolone Glutamate, 17-Hydroxypre Gnanolone and Ganaxolone 

The synergistic antiseizure interaction between cannabinol (100 mg/kg) and neurosteroid ganaxolone or benzodiazepine midazolam provides neuroprotection against adult refractory epilepsy by modulating GABAergic inhibition via binding to the δ and γ subunits of GABARs in a dose–response and time–course manner, respectively [234]. Finally, a recent study in the literature has reported that pregnanolone glutamate and its metabolites, particularly pregnanolone and 17-hydroxypregnanolone, are potent positive modulators of GABARs in the brain, but they also have a moderate inhibitory effect on NMDA receptors, similar to pentobarbital and ketamine, thus influencing locomotor activity and behavior of zebrafish larvae [235].

## 9. Food Nutrients and Neurosteroid Hormones in the Gut–Brain Axis 

In recent years, compelling evidence has highlighted that steroid hormone dysfunction leads to intestinal damage and barrier permeability and, consequently, the onset and progression of CNS disorders, including neurodegenerative and neuropsychiatric disorders, along the gut–brain axis [236]. In this new light, the discovery of targeted nutritional strategies through polyphenols, vitamin D, and/or probiotics is emerging as a promising tool in regulating endogenous neurosteroid changes, with the potential to effectively prevent or mitigate brain and gastrointestinal diseases.

### 9.1. Potential Crosstalk Between Food Nutrients and Neurosteroids for Gut and Brain Health via GABARs

Recent research has widely elucidated the hormetic dose–response effects of functional nutrients, particularly polyphenols alone and/or in synergy with probiotics [237], in preserving endogenous neurosteroid homeostasis through the potential activation of the Nrf2 pathway and GABAergic signaling and the inhibition of oxidative stress and neuroinflammatory pathways, as well as ferroptosis in vitro, in vivo and in humans via the gut–brain axis [237,238]. The functional interplay between gut microbiota and the brain involves multiple cellular pathways and molecules, including the vagus nerve, tryptophan production, extrinsic enteric-associated neurons, LXA4, neurotransmitters (i.e., GABA, serotonin, acetylcholine and dopamine), HPA axis, nutrients, short-chain fatty acids (SCFAs), resilience phase II genes and neurosteroids [238]. Perturbations in the gut microbiota composition and function, termed dysbiosis, disrupt the intrinsic balance between beneficial and pathogenic bacteria, typically favoring the latter [238]. The gut microbiome, profoundly influenced by food nutrients, emerges as a key player. The concept of “microgenderome” evoked by Yoon and Kim and related to the potential role of sex hormones and gender in the modulation of gut microbiota is also documented [239]. This term describes sex differences in bidirectional interactions between the microbiota, circulating steroid hormones, immunity and disease susceptibility [239]. Importantly, the enteric nervous system (ENS) is an intrinsic autonomous network of neuronal ganglia in the intestinal tube with about 100 million neurons located in the myenteric plexus and submucosal plexus. For this reason, it is also termed the “brain within the gut” or “the second brain” [240]. Neurosteroids are well known to influence both the CNS and peripheral nervous system (PNS), exhibiting neuroprotective, neural excitability and neuroplastic effects [241]. The intricate mechanism, by which neurosteroids, particularly progesterone, shields the ENS, remains the subject of ongoing research. Functional food nutrients are the major determinant factors implicated in shaping gut microbiota composition across the lifespan. Importantly, dietary interventions (probiotics) in synergy with neurosteroid-based therapies targeting several mechanisms, including microbial metabolites, neuronal- immune- and metabolic-pathways and neurotransmitters such as GABA, may have the potential for treating neurodegeneration and psychiatric disorders in order to preserve gut and brain health via GABAR activation [241,242,243].

#### 9.1.1. Progesterone 

A recent study has shown, for the first time, that progesterone exerts neuroprotective effects within the ENS [242]. Interestingly, treatment with 10 nM of progesterone significantly improves brain resilience under neuropathological conditions by enhancing progesterone receptors, especially the expression of progesterone receptor membrane component 1 (PGRMC1) of non-genomic signaling pathways, thus mediating the neuroprotective effects in the ENS in neuronal cells and experimental PD models [242]. Additionally, a low dose of 8 mg/kg of progesterone promoted neuroprotection and immunomodulatory activity in the gut after MPTP-induced lesions by upregulating BDNF levels in a mouse model of PD [241]. The ENS, as a part of the PNS, contains several neurotransmitters, including GABA, which is the primary target of neurosteroids. GABA is also a neurotransmitter of enteric interneurons, targeting excitatory GABA_(A)_ or inhibitory GABA_(B)_ receptors that regulate gut motility and mucosal function of the neuroendocrine network [243]. Although basal and peak neurosteroid levels have been detected at nanomolar concentrations under physiological circumstances, these concentrations are sufficient to positively modulate GABARs [244]. Indeed, enteric GABA receptor alterations lead to CNS disorders and can represent potential novel target sites for natural drug development [243,244].

#### 9.1.2. Allopregnanolone 

Numerous evidence has shown that neurosteroids can modulate GABAergic neurotransmission in the ENS, determining changes in gut motility and secretion [245,246]. For example, it has been found that the neurosteroid allopregnanolone enhances GABAergic neurotransmission by binding to GABARs, especially αβδ isoforms in the ENS, leading to increased colonic motility in rodents [245]. In addition, aberrant levels of neurosteroids in fecal samples have also been implicated in the pathogenesis of various gastrointestinal disorders, including IBS and inflammatory bowel disease (IBD) [246]. Unfortunately, research on the interaction between nutrients, gut microbiota and neurosteroids in the context of ENS is limited. Nevertheless, some studies suggest that dietary interventions may affect the gut microbiota and subsequently impact ENS and neurosteroid signaling [247,248,249].

#### 9.1.3. DHEAS

It is well known that physical and psychological stress alters the activity of the gut–brain axis, potentially inducing intestinal barrier dysfunction that can, in turn, influence the release of neurosteroids. This exacerbates neuroinflammation and BBB permeability, and impairs neurotransmission homeostasis, ultimately leading to the development of cognitive and mood disorders. In this scenario, a recent clinical study of 71 male U.S. Marines shows that, after stressful military survival training, increased norepinephrine, epinephrine and DHEA-S levels are associated with an increase of BBB permeability and decrease in body mass. Likewise, the decrease of fat-free mass is associated with decrease in liposaccharide-binding protein (LBP) concentrations. Conversely, increased concentrations of S100 calcium-binding protein B (S100B) are related to a decrease in psychomotor vigilance but not with changes in LBP concentrations [250]. Therefore, physical and psychological stress activates distinct mechanisms leading, on the one hand, to an increase in BBB permeability which is linked to cognitive decline, and, on the other, to fat-free mass loss, which is related to mood disorders.

#### 9.1.4. Probiotics and Medicinal Plants

Probiotic therapy with *L. helveticus* CCFM1076, targeting the gut–brain axis, restored the homeostasis of inhibitory (GABA) and excitatory (glutamate) neurotransmitters by improving the balance of the 5-hydroxytryptamine (5HT) system in the PNS and CNS correlated with increased intestinal SCFAs levels (butyric acid) and decreased *Turicibacter* abundance, thereby ameliorating autistic-like behaviors in rats exposed to valproic acid on day 12.5 of pregnancy [247]. Intriguingly, it has been observed that changes in the expression of GABA_A_ receptor subunits affect depressive and anxiety-like behaviors. Nevertheless, a selection of vital gut bacteria can regulate the function of GABARs. Specifically, *Lactobacillus rhamnosus JB-1* modulated GABA_Aα2_, GABA_Aα1_, and GABA_B1b_ in the brain, which led to a decrease in depression and anxiety via the vagus nerve [248]. Additionally, oral supplementation with *Lactobacillus plantarum SNK12* (4 × 10^8^ cells/head) upregulated the expression of hippocampal neurotrophic factors such as *bdnf* and *nt3* and GABARs in stressed animals [249]. In addition, a recent study has reported the protective effects of glycated milk casein fermented with *Lactobacillus rhamnosus* 4B15 on intestinal and neuronal disorders, such as anxiety-like behaviors, in mice under chronic stress by regulating GABARs through the HPA pathway along the gut–brain axis [251]. Furthermore, a synbiotic (Syn) combination of prebiotics and probiotics composed of mixed oligosaccharides and bacterial colonies, including *Bifidobacterium bifidum*, *Bifidobacterium infantis*, *Bifidobacterium longum*, *Lactobacillus acidophilus*, *Lactobacillus casei,* and *Lactococcus lactis*, and in synergy with venlafaxine (Vlx), improved gut permeability, reduced inflammation and ultimately enhanced depressive-like behaviors and spatial learning–memory impairment in stressed rats. This study has also shown that Syn and Vlx partly contribute to affect the expression of the glial cell-derived neurotrophic factor (GDNF) in the hippocampus and intestine, suggesting particularly protective effects on both the gut barrier and the brain [252]. Likewise, a traditional Chinese medicine termed Qinglong Zhidong Decoction ameliorated tic-like behavior by restoring the balance of gut microbiota and neurotransmitters. Specifically, it increased GABA content and decreased glutamate and dopamine levels in serum and striatum via activation of GABARs in a mouse model of Tourette’s disease [253]. Overall, the data are limited, but current research suggests a potential crosstalk between ENS, neurosteroids and food nutrients along the gut–brain axis via GABARs, opening up novel avenues for promising therapeutic interventions. Personalized nutritional therapy investigating individualized dietary approaches based on microbiome and metabolic phenotypes could revolutionize the prevention and management of nervous system disorders.

## 10. Innovative Technology in the Study of Polyphenols and Neurosteroids in Neurodegeneration and Psychiatric Diseases

Research on cerebral organoids (COs) over the last decade has shown their utility in studying human brain development and function [254]. Derived from human pluripotent stem cells (hiPSCs), these 3D multicellular aggregates can self-organize or be shaped to resemble various CNS regions [255]. COs exhibit functional neuronal receptors and ion channels, forming synapses and intricate signaling pathways [256]. Therefore, COs are suitable for studying psychiatric and neurodegenerative disorders, such as AD [257], PD [258], ASD [259] and depression [260], as well as brain tumors [261].

### 10.1. Neurodegenerative Disorders: AD and PD 

COs developed for neuronal diseases exhibit disease-specific features. For instance, COs from familial AD donors recapitulate the neuropathological phenotype by displaying increased levels of Aβ protein and tau protein [257]. Organoids with the *LRRK2G2019S* and *DNAJC6* mutations show increased quantity of α-synuclein oligomers and heightened cell death, contributing to neuronal GABAR dysfunction and juvenile-onset PD [256,258]. In neurodegenerative diseases, neurosteroidogenesis can be significantly altered. Evidence suggests that neurosteroids, such as 17β-estradiol, DHEA and allopregnanolone, regulate neurodegeneration and neuroinflammation, on the one hand by supporting neuronal survival directly and, on the other, by moderating inflammatory responses of microglia and astrocytes [143]. COs can elucidate the role of neurosteroids in neurodegenerative and psychiatric disorders [256]. For example, COs show downregulation of allopregnanolone, a positive regulator of GABARs in PD. Notably, low doses of 0.5 μM and 1 µM of allopregnanolone improve neuronal communication, possibly through enhanced GABAergic inhibition, suggesting an excitatory/inhibitory imbalance and disrupted neuronal network communication during PD [256]. Because neurosteroids and other molecules cross the BBB, maintaining its structural and functional integrity is crucial for neuronal balance and optimal brain function. BBB breakdown plays a key role in the onset and progression of neurodegenerative diseases [262]. Increased ROS in the brain enhance BBB permeability, causing neuroinflammation and the initiation and progression of neurodegeneration [263]. This oxidative stress damages mitochondria, impairs DNA repair and accelerates neurodegenerative disorders [264]. Therefore, enhancing the consumption of nutritional antioxidants improves the quality of life of patients affected by these pathologies [265].

### 10.2. Autism 

Cortical organoids derived from IPSCs of autistic patients overexpress the FOXG1 transcription factor and increase GABAergic inhibitory neurons, leading to abnormal cell fate, proliferation, shortened cell cycle and an unbalanced inhibitory/excitatory neuronal ratio [259]. Additionally, recent research indicates that dysregulated neurosteroid levels (starting in utero) are linked to imbalances in excitatory/inhibitory neurotransmission and neuroinflammation in ASD. Human PSC-derived 2D and 3D neural models offer a unique chance to study the effects of inflammation on brain development in utero. These models allow us to test the impact of ASD-related genetic mutations on inflammation, preserving the genetic background of patients. Transcriptional studies show these PSC-derived models mimic mid to late fetal brain development stages [266].

### 10.3. Food Nutrients and Organoid Platforms for Modeling Neurological Disorders

Emerging literature indicates that certain polyphenols and their metabolites can cross the BBB, enhancing neuroprotective signaling and neurohormonal effects to provide anti-neuroinflammatory and antioxidant benefits. Polyphenols synthesized by plants with chemical features related to phenolic substances improve brain functions by directly impacting cells and processes in the CNS. Currently, a polyphenol-based nutritional approach using innovative organoid platforms is attracting considerable interest for modeling nervous system disorders in order to promote brain health [267,268,269,270].

#### 10.3.1. Curcumin 

Polyphenolic compounds such as CUR, especially when combined with piperine, act as natural antidepressants by inhibiting MAO-A and MAO-B enzymes, which increases serotonin and dopamine levels in CNS [267]. CUR enhances stress resilience response and protection by restoring hippocampal BDNF and CREB signaling following brain injury [268]. In fact, a low dose of CUR (0.4 mg/kg) promotes hippocampal neurogenesis and cognitive improvement by boosting BDNF levels, thus enhancing memory and learning [268]. In contrast, a high dose of CUR (10 mg/kg) does not effectively upregulate the proliferation or survival of newborn cells [268]. Most recently, Ye et al. have synthesized MM@MnO_2_-Au-mSiO_2_@CUR nanomotors that have the ability to cross the BBB and reach the brain, exhibiting excellent neuroprotective and anti-neuroinflammatory effects by increasing the levels of CAT and SOD as well as TGF-β and IL-10 in vitro, in vivo and in organoids, respectively [269]. Therefore, CUR offers improved mental health by reducing inflammation and oxidative stress, which are key factors in the development of neurodegenerative diseases.

#### 10.3.2. Epigallocathechin-3-Gallate 

A major bioactive constituent of green tea, EGCG, has demonstrated significant neuroprotective properties in diseases like AD and PD. It reduces oxidative stress, scavenges free radicals, chelates metal ions, and inhibits pro-inflammatory pathways, thereby protecting neurons from damage. EGCG also influences cell signaling pathways crucial for neuroprotection and inhibits β-amyloid aggregation in AD, reducing toxic plaque formation [270]. In PD, it protects against the neurotoxins that induce dopaminergic neuron degeneration, acting as a potential therapeutic candidate for neurodegenerative diseases [271]. Interestingly, a recent study has demonstrated that a dose of 50 µM of EGCG, combined with standard chemotherapy, effectively inhibited growth and post-treatment recovery, invasion, and migration of 3D ovarian cancer spheroids and organoids in a dose-dependent manner [272].

#### 10.3.3. Resveratrol

Importantly, RSV at different doses, such as 2 μm/L, 10 μm/L, or 50 μm/L, induced brain health effects on the development of the nervous system and reduced cell apoptosis via SIRT1 pathway activation in hiPSCs-derived COs [273]. Specifically, it inhibits the expression of caspase-3 to protect the development of the nervous system during the embryonic period via SIRT1 activation. Therefore, current research indicates that moderate doses of RSV supplements may have neuroprotective effects on brain development and cognitive behavior during pregnancy or early childhood.

#### 10.3.4. Hidrox^®^

HD may exert therapeutic effects during the pathogenesis of AD due to its potent antioxidant and anti-neuroinflammatory properties, as well as its ability to interfere with Aβ aggregates and tau proteins [274]. Additionally, HD modulates mitochondrial dysfunction, ensuring the proper energy metabolism essential for brain function [274]. More recently, a study by Ercelik et al. revealed that HD (100 µM to 500 µM), in synergy with temozolomide, possesses the ability to inhibit 3D spheroid proliferation by reducing ROS production and tumor cell migration, underlining its potential as an adjuvant treatment in glioblastoma [275].

#### 10.3.5. Sulforaphane

SFN ensures brain health through several protective mechanisms. It activates the Nrf2 pathway to encode phase II detoxification enzymes. It does so by reducing oxidative stress and inhibiting NF-κB to block pro-inflammatory cytokine cascade and neuroinflammation in neurons [276]. In the brain of transgenic mouse models of AD, SFN reduces the levels of aggregated Aβ and phosphorylated tau proteins [276]. Consequently, the multi-targeted modulation of polyphenols and neurosteroids, using promising non-invasive in vitro 3D modeling, may help prevent the progression of neuropsychiatric and neurodegenerative disorders [277,278] as well as brain tumors [279,280], offering a new therapeutic strategy for managing such neuropathophysiological conditions. Taken together, studies using COs could significantly contribute to further elucidate the role of nutritional medicine with polyphenols and their therapeutic potential impact on the action of neurosteroids in order to prevent or delay the development of neurodegenerative and psychiatric disorders.

## 11. Conclusions and Future Perspectives 

In summary, exploring the molecular and cellular mechanisms underlying functional food nutrients, such as polyphenols, more bioavailable polyphenol-combined nanoparticles, as well as probiotics, vitamin D and PUFAs at moderate/low doses (neurohormesis), while targeting the Nrf2 pathway and redox resilience phase II genes and their potential crosstalk with endogenous neurosteroids, in order to regulate their biosynthesis and the physiological levels in the brain during oxidative stress, neuroinflammation, altered neurotransmitters, gut–brain axis dysfunction and BBB permeability, provides opportunities to better target personalized nutritional therapies. Furthermore, this research can broaden our understanding of why neurohormetic nutrients alone and/or in synergy with steroid hormones have neuroactive effects and how they modulate neuronal activity in physiological and neuropathological conditions via GABARs. This is crucial, as recent evidence has highlighted that alterations in neurosteroid synthesis and function have been closely linked to GABAR changes, ultimately leading to the onset and progression of several neurodegenerative and neuropsychiatric disorders, such as AD, PD, ASD and depression. In light of this, vitamin D deficiency is a prominent risk factor for the development of neurological disorders, such ASD and depression, as observed in experimental and clinical studies [116,117,118,124]. However, treatment with vitamin D, in synergy with SFN, provides neuroprotection under neuroinflammatory conditions [90]. This assumes that nutrients and neurosteroids, alone or in synergy, are potent neuromodulators of GABARs and that their neuroprotective and therapeutic potential likely arises from the dose-dependent activation of these endogenous receptors in the brain. In accordance with neurohormesis, the discovery of specific natural formulations with a better (i) quantity and quality of antioxidants, (ii) bioavailability in the intestinal tract, (iii) absorption rates, (iv) hepatic biotransformation and metabolism; (v) pharmacodynamics and target-tissue accumulation and (vi) excretion rates are the main goals of therapeutic nutritional research. The study of the complex interplay between functional food nutrients targeting Nrf2 pathway and GABA signaling via GABARs to modulate circulating neurosteroids, particularly DHEAS, pregnenolone, allopregnanolone and 17β-estradiol, and their derivatives under oxidative stress and neuroinflammation, could substantially benefit from cutting edge “omics” technologies, such as transcriptomics, redoxomics, proteomics, lipidomics and metabolomics. These approaches would reveal, in great detail, the mechanisms regulated by neurosteroids and could therefore offer valuable knowledge for preventing and treating nervous system disorders. Similarly, mapping the presence of polyphenols or their metabolites in human brain tissues through metabolomics is crucial in understanding the degree of absorption, the metabolism, the conjugation and the excretion of polyphenols, which can enhance clinical effectiveness, as significant variability has been found both within and between individuals [281]. Finally, the implementation of innovative human mini-brains (cerebral organoids) for studying oxidative stress, BBB damage and neuroinflammation could comprehensively elucidate the role of polyphenols and their potential impact on the action of neurosteroids, contributing to a deeper understanding of neurohormetic mechanisms and redox resilience/adaptive responses targeting the Nrf2 pathway and GABAergic signaling for the prevention and management of major neurodegenerative and neuropsychiatric disorders, ultimately promoting brain health in humans. Further research in this promising field is clearly warranted to establish the precise cellular mechanisms and molecular targets of personalized nutritional and neuroregenerative therapies, through neurohormetic nutrients alone or in synergistic combination with neurosteroids, and their impact on the brain. Careful evaluation of the appropriate doses to promote neuroprotective actions, with the potential for eventual employment in experimental and clinical settings, is essential.

## Figures and Tables

**Figure 1 ijms-25-12155-f001:**
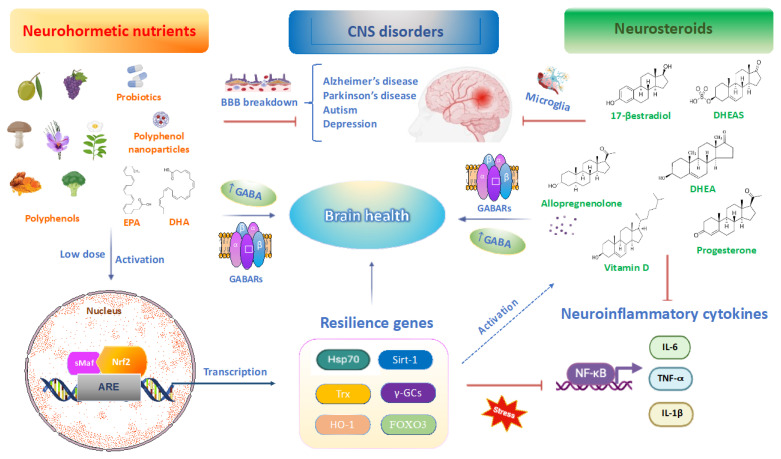
Neurohormetic nutrients and neurosteroids promote brain health through the activation of Nrf2 pathway and GABARs. Neurohormetic nutrients, including polyphenols (resveratrol, Hidrox^®^, sulforaphane, curcumin and 3-epigallocatechin gallate) but also probiotics and ω3 fatty acids (EPA and DHA), protect neurons against oxidative injury and neuroinflammation in a dose-dependent manner. Interestingly, moderate/low doses of food nutrients can modulate the antioxidant pathway and stress resilience genes and proteins, particularly, Hsp70 and HO-1, γ-GCs, Sirt1 and FOXO3, which efficiently remove ROS and provide neuroprotection during CNS disorders. In addition, functional nutrients induce brain health by upregulating GABA via GABARs activation. Likewise, neurosteroids such as vitamin D, 17β-estradiol, DHEA, DHEAS, progesterone and particularly allopregnanolone are potent neuromodulators of GABA through GABARs. The activation of GABARs inhibits neuroinflammatory cytokine cascade and microglia activation, thereby preventing BBB dysfunction and the onset and progression of neurodegenerative and psychiatric disorders. We hypothesize that the synergistic action of neurohormetic nutrients and neurosteroids could potentiate their neuroprotective and neuroregenerative effects through a potential crosstalk between the Nrf2 pathway and GABAergic signaling via GABARs for the prevention and therapy of CNS disorders, ultimately promoting brain health and longevity in humans.

**Figure 2 ijms-25-12155-f002:**
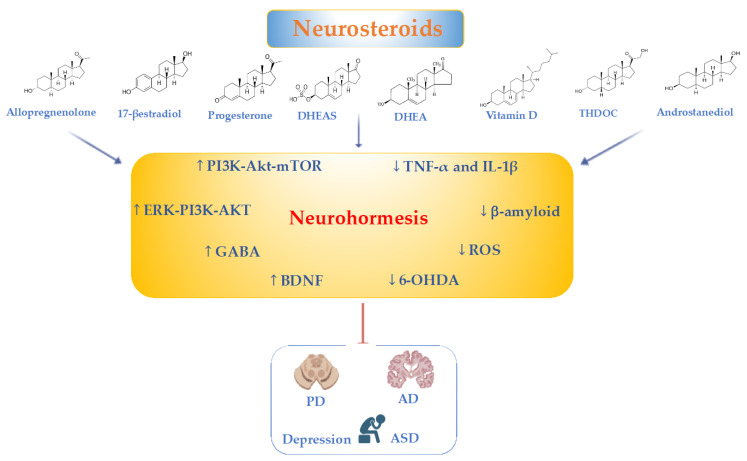
Potential molecular pathways ↑ upregulated or ↓ downregulated by neurosteroids in nervous system disorders in a dose-dependent manner (Neurohormesis).

**Figure 3 ijms-25-12155-f003:**
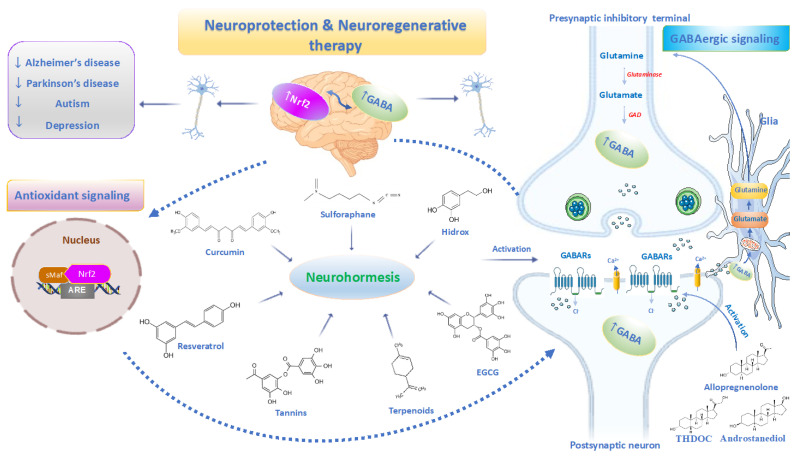
Overview of the potential crosstalk between Nrf2 pathway and GABAergic signaling activated by functional nutrients and neurosteroids through GABARs for neuroprotection and neuro-regenerative therapy via neurohormetic dose–response effects. GAD: glutamic acid decarboxylase. ↑ upregulated and ↓ downregulated.

**Table 1 ijms-25-12155-t001:** Protective mechanisms and molecular pathways upregulated ↑ or ↓ downregulated by resveratrol.

Nutrients	Pathways	Dose	Outcomes	References
	↑ Estradiol and neprilysin	4 mg for 7 days	Reverses cognitive decline and memory, ↓ Aβ deposition in mice.	[62]
	↑ SIRT1	1–10 μM	Promotes healthy ageing through ERα activation and ↑ oestrogenic.	[63]
		50–100 μM	Represses ERα activation and ↑ antioestrogenic.	
RSV	↑ CREB, ↑ StAR ↑ MAPK/ERK 1/2	0.1 to 10 μg/mL	↓ ROS content and ↑ estradiol and progesterone levels. Improves steroidogenesis activity in human granulosa cells dose dependently.	[67]
	↑ caspase 3, ↓ AKT	50–100 μg/mL	↑ ROS content and p27 and p21, ↓ cyclin D2.	
	↓ DHEA ↓ androstenedione ↓ 11-deoxicortisol	10 μM	Inhibits steroidogenesis in adrenocortical cells. The intake of this polyphenol at high doses by women who are at early stages of pregnancy is not recommended.	[69]
	↓ SRD5A1 ↓ AKR1C9 ↓ RDH2	100 μM	Direct inhibitory effects on neurosteroidogenic enzymes such SRD5A1, AKR1C9 and RDH2 to regulate neurosteroids in the rat brain.	[70]
	↑ adrenic acid, docosahexaenoic and eicosapentaenoic acids ↓ F4-neuroprostanes ↓ F2-dihomo-isoprostanes	200 mL	↓ oxidative stress and DNA damage in the CNS and ↑ melatonin and HT after red wine intake in healthy volunteers.	[75]
RSV + L-carnosine	↑ Allopregnanolone and 5α-androstanediol	RSV 20 mg/kg + L-carnosine 200 mg/kg/day	Alleviates the toxic effects of alkylating drugs and improves testis and sperm parameters in rats.	[71]
RSV + Metformin	↓ LH and FSH	RSV 20 mg/kg/+metformin 300 mg/kg/day	Improves hormone profile and ovarian follicular cell architecture in albino rats.	[72]
RSV + Sertraline RSV	↑ SIRT1 and AMPK ↑ Progesterone ↑ allopregnanolone ↑ HPA ↓ androstenedione ↓ DHEA and DHEAS	RSV 20 and 40 mg/kg + sertraline 15 mg/kg 1000 mg	Antagonizes the decrease of progesterone and allopregnanolone levels and attenuates HPA dysfunction by improving behavioral deficits in the prefrontal cortex and hippocampus of stressed mice. ↓ serum levels of the androgen precursors androstenedione, DHEA and DHEAS in middle-aged men with metabolic syndrome after 4 months.	[73] [74]

**Table 2 ijms-25-12155-t002:** Protective mechanisms and molecular pathways upregulated ↑ or ↓ downregulated by curcumin.

Nutrients	Pathways	Dose	Outcomes	References
CUR	↓ TNF-α, IL-β1, IL-6	100 mg daily	Improves cognitive performance and regulates dopamine and norepinephrine levels in different brain areas of ovariectomized rats.	[78]
	↓ TLR4/MyD88/TRAF-6/NF-κB	100 mg/kg once daily	Attenuates boldenone-induced neurobehavioral disturbances, restores antioxidant balance and represses neuroinflammation.	[79]
	↑ GLUT4 and PTEN ↓ Testosterone, IRS1 ↓ PI3K and AKT	200 mg/kg	Modulates serum hormone levels (e.g., 17 β-estradiol, follicle stimulating hormone, luteinizing hormone, progesterone and testosterone in PCOS.	[82]
	↑ SOD, CAT and GPx ↑ PARP ↓ NF-κB	80 mg/kg	Represses oxidative stress through a reduction in lipid peroxidation products and improves semen quality and testosterone hormone levels in rats.	[83]
Curcuméga^®^ + Gabolysat^®^ + Glutamine	↓ CXCL1, TNFα and IL1β ↑ IL-10	Curcuméga^®^ 500 or 100 mg/kg + Gabolysat^®^ 30 mg/kg by oral gavage	Reduces gut barrier disruption and inflammatory responses in murine models of IBS.	[80]
CUR + Lycopene	↓ testosterone, DHT and E2 ↓ IL-1β, IL-6 and TNF-α	CUR 2.4 mg/kg + lycopene 12.5 mg/kg	Attenuates benign prostate hyperplasia and inflammatory process in vivo.	[81]
CUR + Placebo	↓DHEA ↓FPG	500 mg 3 times daily	Improves hyperandrogenemia and hyperglycemia in patients with PCOS.	[84]
CUR + Teopolioside	Not specified	CUR 75 mg + teopolioside 35 mg	Enhances symptoms associated with hyperandrogenism in women with PCOS after 12-weeks.	[85]

**Table 3 ijms-25-12155-t003:** Protective mechanisms and molecular pathways upregulated ↑ or ↓ downregulated by sulforaphane.

Nutrients	Pathways	Dose	Outcomes	References
SFN	↑ 3α-HSD ↑ DHT ↓ testosterone	10 mg/kg	Enhances 3α-HSDs in the liver and the degradation of DHT to block androgenic alopecia in murine hepatocytes and in rodents.	[88]
SFN + 17β estradiol	↑ Nrf2 ↑ GSH, HO-1, SOD, CAT, Trx and NQO1	10–50 nM	Inhibits oxidative damage, ROS production and 8-OHdG levels in cardiomyocytes.	[89]
SFN + honokiol	↑ Nrf2 ↑ Sirt1	1 μM	Regulates the oxidant/antioxidant environment for the control of testosterone homeostasis in aging Leydig cells.	[90]
SFN + Vitamin D	↑ MAPK/ERK ↓ VEGF ↓ TGF-β	1 or 10 μM	Neuroprotection and antioxidant and anti-inflammatory effects in age-related macular degeneration in vitro.	[91]
	↑ JNK/MAPK ↑ Nrf2 ↑ Bax ↓ Bcl-2	SFN 4, and 8 µM + Vitamin D 16 nM	Induces cytotoxicity by activating oxidative stress, DNA damage, and autophagy in prostate cancer in vitro	[92]

**Table 4 ijms-25-12155-t004:** Protective mechanisms and molecular pathways upregulated ↑ or ↓ downregulated by Hidrox^®^.

Nutrients	Pathways	Dose	Outcomes	References
	↑ Nrf2 ↑ HO-1 ↑ GSH, SOD and catalase ↓ NF-κB	10 mg/kg	Modulates oxidative stress and neuroinflammation in the bladder and spinal cord in order to prevent or slow AD and PD in rodents.	[95,96,97]
HD	↑ Nrf2 ↑ GSH, HO-1, SOD, CAT, Trx and NQO1	250 mg/kg	Neuroprotective action, ↑ lifespan and stress resistance and ↓ neurotoxic aggregates of misfolded α-synuclein in dopaminergic neurons of transgenic PD models.	[98,99]
	Not specified	3 g twice daily	Enhances cognitive function, specifically memory, attention, reaction time and executive function in middle-aged and older adults after 12 weeks.	[101]
	↑ p-ERK, PKA, p-AKT, and ZAG	20 mg/kg	↑ plasma testosterone and its metabolite testosterone glucuronide, ↑ L-carnitine and its propionyl-L-carnitine, ↑ beneficial gut bacteria and ↓ bile acids and improves spermatogenesis and semen quality after 2 months.	[102]

**Table 5 ijms-25-12155-t005:** Protective mechanisms and molecular pathways upregulated ↑ or ↓ downregulated by EGCG.

Nutrients	Pathways	Dose	Outcomes	References
EGCG	↓ PKA/PKC ↓ P450scc	20 μg/mL	Reverses the inhibitory effect on 22(R)-hydroxycholesterol, androstenedione and P450scc function, suggesting that inhibition is dose-dependent.	[104]
		100 μg/mL	Blocks testosterone release in rat Leydig cells. Regulates 17β-HSD and leads to the reduction of cellular steroidogenic capacity.	
	↓ 11β-HSD1	25–100 µM	Inhibits the cortisol producing enzyme 11β-HSD1 dose dependently in vitro and in silico.	[107]
	↑ GABA	50–200 μg i.c.v.	Attenuates stress behavior and plasma corticosterone concentration by inducing anti-anxiety, sedative and hypnotic effects in a dose-dependent manner.	[108]
	↑ PKA/CREB	5 μM and 10 μM	↑ StAR expression and progesterone production in human granulosa cells.	[109]
	↑ SOD ↓ IL-6 and TNF-α	25 mg	↑ Antioxidant defense system to block oxidative stress and neuroinflammatory response in aged albino rats.	[110]
	↑ DHEA	100 mg	Strengthens systemic immunity by enhancing cellular immune response in animals.	
EGCG + Caffeine	Not specified	EGCG 0.1 and 0.2 mM + caffeine 10–30 mM	↓ total lipids, triglycerides and cholesterol in C. elegans dose-dependently.	[111]
EGCG + Vitamin D	↓ AGEs	EGCG 120 μM + vitamin D 0.1 μM	Exerts antiglycation ability by decreasing AGEs, ROS overproduction, DNA damage and cytotoxicity in PCOS in vitro and in silico.	[112]
EGCG + ferulic acid	↑ SOD1, GPx1 ↑ ADAM10 ↑ α-secretase ↓ β-secretase ↓ BACE1 ↓ TNF-α and IL-1β	30 mg/kg each once daily p.o.	Inhibits oxidative stress and neuroinflammation, reverts cognitive impairment and mitigates synaptotoxicity and β-amyloid deposits in AD transgenic mice after 3 months.	[113]
EGCG + vitamin D + D-chiro-inositol	Not specified	EGCG 300 mg + vitamin D 50 mg + D-chiro-inositol 50 mg in 2 pills daily	↓ the time required to perform surgery and bleeding during surgery without affecting liver function after 3 months in women with uterine fibroids.	[114]

**Table 6 ijms-25-12155-t006:** Protective mechanisms and molecular pathways upregulated ↑ or ↓ downregulated by vitamin D.

Nutrients	Pathways	Dose	Outcomes	References
Vitamin D	↑ Nrf2, HO-1, Sirt1 ↓ NF-κB, TNF-α, IL-1β	100 μg/kg	Neuroprotection and improved neuronal synapse and memory, was found to abrogate Aβ amyloid production in a rodent model of AD after 4 weeks.	[122]
Vitamin D + Prednisolone	↑ CYP24A1 ↑ IκB ↓ NF-κB	Vitamin D 1000 IU/kg + prednisolone 5 mg/kg	Prevents and regulates prednisolone-induced neurotoxicity and behavioral disturbances by reducing ROS and inflammation, exerting antidepressant-like effects in rats after 30 days.	[123]
Vitamin D + Placebo	↓ IL-1β, IL-6 and hs-CRP	50,000 IU 2 wks^−1^	Improves depressive symptoms in patients after eight weeks.	[124]

**Table 7 ijms-25-12155-t007:** Protective mechanisms and molecular pathways upregulated ↑ or ↓ downregulated by omega-3.

Nutrients	Pathways	Dose	Outcomes	References
Omega-3	↑ DHP ↑ THDOC ↑ isopregnanolone	DHA % 22:6n-3 EPA % 20:5n-3	DHA deficiency contributes to HPA axis hyperactivity. ↓ DHA correlate with ↑ DHP levels in psychiatric patients, while ↓ DHA correlate ↑ THDOC and isopregnanolone levels in healthy subjects.	[127]
	↑ BDNF ↓ TXB	2400 mg	Enhances BDNF and reduces TXB levels in children and adolescents with depressive disorder after 12 weeks.	[129]
	↑ KYN/TRP ↑ SER	2400 mg	↑ kynurenine/tryptophan ratio in depressed children and adolescents after 12-weeks.	[130]
Omega-3 + Vitamin D	Not specified	Omega-3 750 mg + vitamin D 2000 IU capsules per day	Improves core symptoms in children with ASD after a 24-month follow-up period.	[131]

**Table 8 ijms-25-12155-t008:** Protective mechanisms and molecular pathways upregulated ↑ or ↓ downregulated by polyphenol-based nanoparticles.

Nutrients	Pathways	Dose	Outcomes	References
RSV-loaded with gold nanoparticles	↑ 3α-HSD ↑ DHT ↓ testosterone	10 mg/kg	Prevents 17β-estradiol/ERα-induced neuroglobin accumulation and induced apoptosis in cancer cells.	[136]
Zinc oxide–RSV nanoparticles	↑ SOD ↓ MAD	10–50 nM	Attenuates the harmful side effects of levofloxacin in rats.	[137]
CUR-loaded with T807-modified nanoparticles	↑ Nrf2 ↑ Sirt1	5 mg/kg	Crosses the BBB by improving its permeation into the brain. ↓ tau protein and apoptosis in neurons and in vivo.	[138]
CUR-loaded lipid–core nanocapsules	↓ TNF-α, IL-6, IL-1β and IFN-γ	1 or 10 mg/kg	Neuroprotection by reducing Aβ1-42-induced inflammatory cytokines in serum and in the prefrontal cortex and hippocampus of aged mice.	[139]
Theracurmin^®^ + Placebo	Not specified	90 mg twice daily	↑ memory and attention in middle-aged and non-demented adults. Prevents or delays AD progression in non-demented middle-aged and older adults after 18 months.	[140]
CurQfen^®^	↑ BDNF ↓ IL-6 ↓ TNF-α	400 mg × 2/day	Enhances BBB permeability and brain bioavailability. Attenuates AD progression and improves locomotor and cognitive functions in patients with moderate dementia after 6 months.	[141]

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
