# Peer review of "Functional Food Nutrients, Redox Resilience Signaling and Neurosteroids for Brain Health"

_ijms, 2024, doi:10.3390/ijms252212155_

Round 1

Reviewer 1 Report

Comments and Suggestions for Authors

The review by Scuto et al. offers a thorough examination of functional food nutrients and their influence on neurosteroid homeostasis and the Nrf2 pathway. While the content is interesting for audience, the manuscript would benefit from substantial revisions to enhance its readability. Below are some specific suggestions to improve the overall quality:

First, the manuscript is quite lengthy, spanning 37 pages with 284 references. This might overwhelm readers. I recommend streamlining the content by merging similar sections or omitting less critical details. If there is too much material to condense effectively, you might consider splitting the review into two separate papers.

Additionally, many of the paragraphs are very long, which can make the text hard to navigate. Breaking them up into shorter, more concise paragraphs would greatly improve readability.

The manuscript currently reads more like a doctoral thesis than a review article. I strongly recommend that the author follow the specific formatting, citation, and presentation guidelines of the International Journal of Molecular Sciences(IJMS) for preparing Review Articles.

Comments on the Quality of English Language

The manuscript contains numerous grammatical errors, primarily involving the use of commas. I suggest the authors carefully proofread the text to correct these errors and ensure clarity.

Author Response

Response to Reviewer 1

The review by Scuto et al. offers a thorough examination of functional food nutrients and their influence on neurosteroid homeostasis and the Nrf2 pathway. While the content is interesting for audience, the manuscript would benefit from substantial revisions to enhance its readability. Below are some specific suggestions to improve the overall quality:

First, the manuscript is quite lengthy, spanning 37 pages with 284 references. This might overwhelm readers. I recommend streamlining the content by merging similar sections or omitting less critical details. If there is too much material to condense effectively, you might consider splitting the review into two separate papers. Additionally, many of the paragraphs are very long, which can make the text hard to navigate. Breaking them up into shorter, more concise paragraphs would greatly improve readability. The manuscript currently reads more like a doctoral thesis than a review article. I strongly recommend that the author follow the specific formatting, citation, and presentation guidelines of the International Journal of Molecular Sciences (IJMS) for preparing Review Articles.

Response: Thank you very much for taking the time to review this manuscript. The review is thorough as the topic is very complex and the authors want to make readers understand the importance of investigating nutritional medicine with polyphenols, polyphenol-nanoparticles, vitamin D, omega-3 fatty acids, and probiotics alone or in synergy with neurosteroids for targeted preventive and therapeutic interventions in brain disorders.

According to the reviewer’s suggestions, the authors have divided some paragraphs into subparagraphs to improve readability.

  • Neurosteroids as modulators of neuroinflammation
  • 1. In vitro studies
  • 2. In vivo studies
  • The interaction of food nutrients and neurosteroids via GABARs
  • 1. Flavonoids
  • 1.1. Quercetin, apigenin and genistein
  • 1.2. Chrysin
  • 4. Neurosteroids
  • 4.1. Allopregnenolone, THDOC and androstanediol
  • 4.2. Pregnenolone sulfate, DHEA and DHEAS
  • 4.3. Pregnanolone glutamate, 17-hydroxypregnanolone and ganaxolone
  • Food nutrients and neurosteroid hormones in gut-brain axis
  • 1. Potential crosstalk between Food nutrients and Neurosteroids for gut and brain health via GABARs
  • 1.1. Progesterone
  • 1.2. Allopregnanolone
  • 1.3. DHEAS
  • 1.4. Probiotics and medicinal plants

Reviewer 2 Report

Comments and Suggestions for Authors

The research article " Functional food nutrients, redox resilience signaling and neurosteroids for brain health " is a well-researched review; it includes several references to functional nutrients, neurosteroids, and their effects on brain health. The coverage of polyphenols, flavonoids, vitamins, and omega-3 and omega-6 fatty acids is comprehensive and well-supported by scientific evidence. This is an important area of Molecular Endocrinology and Metabolism and will be interesting for the readership of the “ijms”.

The organization of the topics is clear, covering neurosteroidogenesis, functional nutrients, and their impacts on diseases such as Alzheimer's , neurological disorders and depression. The innovative approach of combining functional nutrients with neurosteroids to improve brain resilience explores a promising and emerging area of ​​research.

 Authors are encouraged to consider the following comment and suggestion to further refine their work.

ü  While functional nutrients and neurosteroids are discussed in detail, it would be helpful to provide more information on the molecular mechanisms by which these elements influence specific neuronal signaling pathways.

ü  Also, some concepts, such as the activation of the Nrf2 pathway and the neuroprotective effects of polyphenols are repeated several times. It is preferable to reduce redundancy.

ü  To improve both the understanding and the visual aspect of the review, I propose to add figures and tables to summarize the nutrients or molecular pathways.

ü  I also recommend reviewing the text writing for spelling errors, to double check the references and to improve the iThenticate report.

After making these changes the manuscript can be accepted.

Author Response

Response to Reviewer 2

The research ar)cle " Func)onal food nutrients, redox resilience signaling and neurosteroids for

brain health " is a well-researched review; it includes several references to func)onal nutrients,

neurosteroids, and their effects on brain health. The coverage of polyphenols, flavonoids, vitamins,

and omega-3 and omega-6 faDy acids is comprehensive and well-supported by scien)fic evidence.

This is an important area of Molecular Endocrinology and Metabolism and will be interes)ng for the

readership of the “IJMS”. The organiza)on of the topics is clear, covering neurosteroidogenesis,

func)onal nutrients, and their impacts on diseases such as Alzheimer's, neurological disorders and

depression. The innova)ve approach of combining func)onal nutrients with neurosteroids to

improve brain resilience explores a promising and emerging area of research.

Authors are encouraged to consider the following comment and sugges)on to further refine their

work.

- While func)onal nutrients and neurosteroids are discussed in detail, it would be helpful to provide

more informa)on on the molecular mechanisms by which these elements influence specific

neuronal signaling pathways.

- To improve both the understanding and the visual aspect of the review, I propose to add figures

and tables to summarize the nutrients or molecular pathways.

Response: Authors acknowledge the reviewer for the construc)ve comments. Tables (1-8) and 1

figure have been included to provide more details on the molecular mechanisms and pathways

upregulated or downregulated by func)onal food nutrients.

- Also, some concepts, such as the ac)va)on of the Nrf2 pathway and the neuroprotec)ve effects of

polyphenols are repeated several )mes. It is preferable to reduce redundancy.

Response: Authors acknowledge the reviewer for the construc)ve review. The manuscript focuses

on the Nrf2 pathway and neuroprotec)on by polyphenols alone or in synergy with neurosteroids;

Redundancy has been reduced.

- I also recommend reviewing the text wri)ng for spelling errors, to double check the references and

to improve the iThen)cate report.

Response: The authors thank the reviewer for this comment. The English has been completely revised by a specialist in scien)fic English.

Round 2

Reviewer 1 Report

Comments and Suggestions for Authors

Dear authors,

Thank you for the revised manuscript. I appreciate the improvements made, especially the division into subsections, which enhances the structure. However, the paragraphs remain quite lengthy, which could make the text challenging for readers to navigate. As mentioned in my previous report, breaking these into shorter, more concise paragraphs would significantly improve readability.

If the editors agree with these suggested changes, I believe the manuscript will be in a more suitable form for publication.

Author Response

Response to Reviewer 1

Thank you for the revised manuscript. I appreciate the improvements made, especially the division into subsections, which enhances the structure. However, the paragraphs remain quite lengthy, which could make the text challenging for readers to navigate. As mentioned in my previous report, breaking these into shorter, more concise paragraphs would significantly improve readability.

If the editors agree with these suggested changes, I believe the manuscript will be in a more suitable form for publication.

Response: The authors thank the reviewer for this comment. Based on the reviewer's suggestions, the authors have split other paragraphs and subparagraphs to improve readability. In particular:

  1. Polyphenol-nanoparticle delivery systems and neurosteroid signaling

5.1. Preclinical studies

5.2. Clinical studies

  1. The role of neurosteroids in nervous system disorders: focus on nutrients

7.1. Alzheimer’s disease

7.1.1. Preclinical studies

7.1.2. Clinical studies

7.2. Parkinson’s disease

7.2.1. Preclinical studies

7.2.2. Clinical studies

7.3. Depression

7.3.1. Preclinical studies

7.3.2. Clinical studies

7.4. Autism spectrum disorder

7.4.1. Preclinical studies

7.4.2. Clinical studies

  1. Innovative technology in the study of polyphenols and neurosteroids in neurodegeneration and psychiatric diseases

10.1. Neurodegenerative disorders: AD and PD  

10.2. Autism

10.3. Food nutrients and organoid platform for modeling neurological disorders

10.3.1. Curcumin

10.3.2.  Epigallocathechin-3-gallate 

10.3.3. Resveratrol

10.3.4. Hidrox®

10.3.5. Sulforaphane
